# Functional Proteomics Characterization of the Role of SPRYD7 in Colorectal Cancer Progression and Metastasis

**DOI:** 10.3390/cells12212548

**Published:** 2023-10-31

**Authors:** Ana Montero-Calle, Sofía Jiménez de Ocaña, Ruth Benavente-Naranjo, Raquel Rejas-González, Rubén A. Bartolomé, Javier Martínez-Useros, Rodrigo Sanz, Jana Dziaková, María Jesús Fernández-Aceñero, Marta Mendiola, José Ignacio Casal, Alberto Peláez-García, Rodrigo Barderas

**Affiliations:** 1Chronic Disease Programme (UFIEC), Instituto de Salud Carlos III, 28220 Madrid, Spain; sofiaj01@ucm.es (S.J.d.O.); ruthbena@ucm.es (R.B.-N.); raquel.rejas@isciii.es (R.R.-G.); 2Centro de Investigaciones Biológicas Margarita Salas, CSIC, 28040 Madrid, Spain; icasal@cib.csic.es; 3Translational Oncology Division, OncoHealth Institute, Health Research Institute-University Hospital Fundación Jiménez Díaz-Universidad Autónoma de Madrid, 28040 Madrid, Spain; javier.museros@quironsalud.es; 4Surgical Digestive Department, Hospital Universitario Clínico San Carlos, 28040 Madrid, Spain; rsanzl@salud.madrid.org (R.S.); jana.dziakova@salud.madrid.org (J.D.); 5Surgical Pathology Department, Hospital Universitario Clínico San Carlos, 28040 Madrid, Spain; jmariajesus.fernandez@salud.madrid.org; 6Molecular Pathology and Therapeutic Targets Group, La Paz University Hospital (IdiPAZ), 28046 Madrid, Spain; marta.mendiola@salud.madrid.org (M.M.); alberto.pelaez@idipaz.es (A.P.-G.)

**Keywords:** colorectal cancer, SPRYD7, cancer metastasis, proteomics, protein dysregulation, interactome

## Abstract

SPRY domain-containing protein 7 (SPRYD7) is a barely known protein identified via spatial proteomics as being upregulated in highly metastatic-to-liver KM12SM colorectal cancer (CRC) cells in comparison to its isogenic poorly metastatic KM12C CRC cells. Here, we aimed to analyze SPRYD7’s role in CRC via functional proteomics. Through immunohistochemistry, the overexpression of SPRYD7 was observed to be associated with the poor survival of CRC patients and with an aggressive and metastatic phenotype. Stable SPRYD7 overexpression was performed in KM12C and SW480 poorly metastatic CRC cells and in their isogenic highly metastatic-to-liver-KM12SM-and-to-lymph-nodes SW620 CRC cells, respectively. Upon upregulation of SPRYD7, in vitro and in vivo functional assays confirmed a key role of SPRYD7 in the invasion and migration of CRC cells and in liver homing and tumor growth. Additionally, transient siRNA SPRYD7 silencing allowed us to confirm in vitro functional results. Furthermore, SPRYD7 was observed as an inductor of angiogenesis. In addition, the dysregulated SPRYD7-associated proteome and SPRYD7 interactors were elucidated via 10-plex TMT quantitative proteins, immunoproteomics, and bioinformatics. After WB validation, the biological pathways associated with the stable overexpression of SPRYD7 were visualized. In conclusion, it was demonstrated here that SPRYD7 is a novel protein associated with CRC progression and metastasis. Thus, SPRYD7 and its interactors might be of relevance in identifying novel therapeutic targets for advanced CRC.

## 1. Introduction

Colorectal cancer (CRC) develops slowly in a complex process that usually takes several decades whereby mutated epithelial cells proliferate, grow aberrantly, and increase their capacity to metastasize and colonize neighboring or distal organs [1,2]. During carcinogenesis, cancer cells suffer from metabolic, behavioral, and morphological changes, as well as alterations in their molecular mechanisms. These alterations improve the tumorigenic (proliferation or colony formation) and metastatic (invasion, angiogenesis, migration, or adhesion) properties of cancer cells [3,4]. These changes, associated with the accumulation of genetic, epigenetic, and proteomics variations in tumoral cells, such as mutations in oncogenes and tumor suppressor genes (e.g., *APC* or *TP53* genes), epigenetic silencing of MLH1, or the dysregulation of cytokines or metalloproteinases levels, result in alterations in the expression, localization, or function of different proteins. Some of them have already been identified and associated with CRC progression and metastasis via the disturbance of biological processes, such as Wnt, p53, or TGFβ1 signaling pathways [5,6,7,8,9,10]. However, colorectal carcinogenesis and metastatic processes are far from fully elucidated, and further research is needed in order to attain a better understanding of the molecular pathways and biological processes underlying the development, progression, and metastasis of CRC, which might aid in the identification of novel diagnostic and prognostic biomarkers and therapeutic targets of intervention.

Metastasis is the main reason for about 90% of CRC-related deaths. Therefore, the analysis of dysregulated proteins associated with the metastatic process is of high interest in the identification of novel therapeutic targets that could improve the survival rate of CRC patients. Metastasis is a sequential multi-step process whereby cancerous cells need to interact with different microenvironments in order to migrate from the primary tumor to the metastatic niche after intravasation into the vasculature, survival in circulation, extravasation, and colonization of the secondary organs [10,11]. Thus, the study of the proteome associated with CRC metastasis might reflect the mechanical features and properties that cancer cells must modify in order to adapt to, and survive in, each microenvironment during migration. Most proteomics analyses related to cancer metastasis are based on cellular models resembling critical steps of cancer metastasis [12], thus allowing for the analysis of this process. In this sense, the KM12 CRC cell model of metastasis has been widely used, as the comparison of isogenic cell lines with different metastatic capabilities specifically allows for the analysis of late metastatic events in CRC, including survival and colonization [13,14,15,16]. This model includes the poorly metastatic KM12C cells derived from a CRC patient at Dukes B stage (actual stage II, T3N0M0) and its corresponding isogenic KM12SM highly metastatic-to-liver CRC cells. This model recapitulates quite well features observed in actual CRC patients, such as the dysregulation of key molecules in CRC metastasis such as CDH17, ERBB2, AIP, IL13Rα2, PTPRN, or MMP7, among others [13,14,15,17,18].

In this context, a recent multi-dimensional proteomics study revealed 1318 proteins differentially expressed in diverse subcellular compartments (cytoplasm, membranes, nucleus, chromatin-bound proteins, and cytoskeleton) of highly metastatic KM12SM cells in comparison to the isogenic poorly metastatic KM12C CRC cells [15]. In this work, we have focused on the functional analysis of a barely studied protein among those dysregulated proteins associated with CRC metastasis: SPRYD7 (SPRY domain-containing protein 7) [15,19], with the major aim of elucidating its role in CRC progression and metastasis. SPRYD7 was found upregulated among the membrane subcellular organelles of highly metastatic-to-liver KM12SM cells in comparison to poorly metastatic KM12C CRC cells [15]. Remarkably, this barely studied protein has not been previously associated with CRC, highlighting the novelty and relevance of an in-depth functional proteomics analysis of the molecular and biological pathways directly or indirectly regulated by SPRYD7 in CRC. Here, SPRYD7 was overexpressed in KM12C, KM12SM, SW480, and SW620 CRC cell lines with different metastatic abilities, and the tumorigenic and metastatic changes induced by its stable overexpression have been investigated via in vitro assays to analyze its role in CRC progression and metastasis. These functional cell-based alterations were confirmed in vitro via transient siRNA SPRYD7 silencing. Additionally, in vivo assays were performed with stable KM12C and KM12SM, overexpressing SPRYD7, and control cells. The rationale behind the overexpression of SPRYD7 in CRC cells was to try to determine whether SPRYD7 overexpression would increase the tumorigenic and metastatic properties of KM12C, KM12SM, SW480, and SW620 cells, whereas the rationale behind the use of KM12C and KM12SM CRC cells for in vivo assays was to survey changes in liver homing metastasis and tumor growth and to determine whether SPRYD7 could induce poorly metastatic cells to become metastatic to the liver. Finally, our analyses also included an in-depth proteomics workflow for the identification of the interactome of SPRYD7 and the biological processes and functions altered by its dysregulation in CRC metastasis.

## 2. Materials and Methods

### 2.1. Human Samples

The Institutional Ethical Review Boards of the Instituto de Salud Carlos III and Hospital Clínico San Carlos, Madrid, approved this study on biomarker discovery and validation (CEI PI 45). The Ethical Review Board of Hospital Clínico San Carlos (IdISSC) biobank, which belongs to the National Biobank Net (ISCIII), cofounded with FEDER funds, approved this study prior to obtaining paired tumoral and non-tumoral optimal cutting-temperature (OCT)-embedded tissue samples from CRC patients. Written informed consent was obtained from all patients. Tissue samples were collected using a standardized sample collection protocol and stored at −80 °C until use [13,20].

For RNA extraction, paired tumoral and non-tumoral OCT-embedded tissue samples from 7 CRC patients at stage I (n = 1), II (n = 2), III (n = 2), and IV (n = 2) were used (Appendix A).

For the in silico prognostic value analysis of SPRYD7, the GSE17538 database, containing 244 tumour samples with clinicopathological description from colorectal cancer patients, was used as it has been previously [14]. After data normalization with Bioconductor’s Affymetrix, Kaplan–Meier curves were obtained to assess the prognostic value of SPRYD7. The best cut-off to separate high- and low-expression populations was used. The significance of the difference in survival between both populations was estimated via a log-rank test.

### 2.2. Cell Lines

Two isogenic cellular models of CRC—same genetic background but different metastatic properties—were used for the analysis of the role of SPRYD7 in CRC progression and metastasis. Isogenic KM12C CRC cells with poor metastatic ability, KM12SM CRC cells with high metastatic ability to liver, and KM12L4a CRC cells with high metastatic ability to liver and lung were obtained from the I. Fidler’s laboratory (MD Anderson Cancer Center). Isogenic SW480, with non-metastatic capacity, and SW620 cells, with high metastatic capacity to lymph nodes, were obtained from the American Type Culture Collection (ATCC, Manassas, VA, USA) cell repository. HT-29, Caco-2, RKO, SW48, and Lim-1215 CRC cell lines were also obtained from the ATCC cell repository. Cells were routinely tested for mycoplasma.

CRC cells were grown at 37 °C and 5% CO_2_ in Dulbecco’s Modified Eagle Medium (DMEM, Lonza) containing 10% fetal bovine serum (FBS, Sigma-Aldrich, St. Louis, MO, USA), 1x L-glutamine (Lonza), and 1x penicillin/streptomycin (Lonza, Basel, Switzerland) (complete medium). Caco-2 were grown in minimum essential medium (MEM, Lonza), supplemented with 10% FBS, 1x L-glutamine, and 1x penicillin/streptomycin (complete medium), at 37 °C and 5% CO_2_.

### 2.3. Cloning, siRNAs, and Transfection

SPRYD7 full-length reference CDS was obtained from the DNASU repository in the flexible pDONR221 vector (HsCD00862560). Then, the SPRYD7 gene was cloned into the pcDNA3.1(+) vector using the NEBuilder HiFi DNA assembly Master Mix kit (New England BioLabs, Ipswich, MA, USA), following the manufacturer’s instructions. Briefly, 1–5 ng of each vector was amplified via PCR with the Advantage 2 polymerase kit (Takara, Kusatsu, Shiga, Japan), using specific oligonucleotides (Appendix A). Then, 15–25 base pair (bp) overlapping DNA fragments was digested with *Dpn*I (Thermo Fisher Scientific, Waltham, MA, USA) and purified via ethanol precipitation. Then, 100 ng of the linearized amplified by PCR pcDNA3.1(+) vector, 2-fold molar of SPYRD7 gene, and 10 µL of NEBuilder HiFi DNA assembly master mix were incubated together in a final volume of 20 µL with PCR water at 50 °C for 60 min for the assembly of DNA fragments. Then, 50 µL of TOP10 *Escherichia coli* (*E. coli*) chemically competent bacterial cells were transformed with 2.5 µL of the assembly reaction, and plasmids were isolated and purified with the Miniprep kit (Neo-biotech, Nanterre, France), following the manufacturer’s instructions. Finally, the quality of purified plasmid was analyzed via agarose gel electrophoresis, and plasmids were sequenced prior to use.

For transfection, 2.5 × 10^5^ KM12C, KM12SM, SW480, or SW620 cells were seeded per well on 6-well plates (corning) in 2 mL complete DMEM medium and transfected using the JetPrime transfection reagent (Polyplus, Illkirch-Graffenstaden, France). Briefly, 2 µg of SPRYD7 or empty (mock-stably transfected) pcDNA3.1(+) vectors were diluted in 200 µL of JetPRIME buffer and incubated with 4 µL of JetPRIME Transfection reagent for 10 min at room temperature (RT). Then, CRC cells were incubated with the reaction solution at 37 °C and 5% CO_2_ during 48 h. Finally, the medium was replaced by complete medium containing 1 mg/mL G418 (Geneticin 418, Santa Cruz Biotechnology, Dallas, TX, USA) for selection (3–4 weeks). After selection, transfected cells were grown in complete medium supplemented with 0.6 mg/mL G418 to establish genetically modified CRC cell lines.

SPRYD7 and control siRNAs were purchased from Sigma (EHU133421 and SIC001, respectively). For siRNA transfections, 2.5 × 10^5^ cells were seeded in culture plates and maintained in DMEM with 10% fetal bovine serum at 37 °C in 5% CO_2_ for 24 h, according to established protocols [18,21]. Then, cells were transfected with 22 pmol siRNA using 2 µL JetPrime Transfection reagent in 100 µL of JetPrime buffer. Then, 24 h after transfection, SPRYD7 silencing was analyzed via Western blot, qPCR, and semi-quantitative PCR [22].

### 2.4. Protein Extraction

Cells at 90% confluence were harvested with 4 mM EDTA (Carl Roth) PBS 1x, centrifuged 5 min at 260× *g* and RT, and lysed with 500 µL of RIPA buffer (Sigma-Aldrich) supplemented with 1x protease and phosphatase inhibitors (MedChemExpress, Monmouth Junction, NJ, USA). Cells were manually disaggregated using 16 G and 18 G needle syringes until homogeneity was observed. Finally, samples were centrifuged at 10,000× *g* and 4 °C for 10 min, and the supernatant (protein extracts) collected in a new tube and stored at −80 °C until used.

The tryptophan quantification method [23] was used to determine the concentration of the cell protein extracts. Coomassie blue staining after 10% SDS-PAGE under reducing conditions was performed as quality control (Appendix A).

### 2.5. Western Blot

For Western blot (WB) analyses, 10 µg of each protein extract were used. Proteins were separated on discontinuous 10% or 15% SDS-PAGE under reducing conditions prior to the transference to a nitrocellulose membrane (Hybond-C extra, GE Healthcare, Chicago, IL, USA) at 100 V for 90 min. Prior to the incubation of the membranes with the corresponding primary antibodies at optimized dilutions (Appendix A) in blocking buffer (3% skimmed milk or 3% bovine serum albumin (BSA) in 0.1% Tween-20 PBS 1x) overnight (O/N) at 4 °C and in rotation, membranes were blocked with blocking buffer for 1 h at RT. After washing 3 times with washing buffer (0.1% Tween-20 PBS 1x), membranes were incubated with the corresponding secondary antibody (Appendix A) diluted in blocking buffer for 1 h at RT in rotation. Finally, membranes were washed 3 times with washing buffer, and the signal was developed with the SuperSignal West Pico Maximum Sensitivity Substrate (Thermo Fisher Scientific) and detected on an Amersham Imager 680 (GE Healthcare). RhoGDI and GAPDH were used as loading controls and for normalization after relative quantification of protein band intensities using the ImageJ software. After transference, membranes were incubated with Ponceau red staining solution as quality control of the total protein content per lane (Appendix A).

### 2.6. RNA Extraction, cDNA Synthesis, PCR, and qPCR

For RNA extraction, cells at 90% confluence were harvested with trypsin-EDTA (Lonza) and centrifuged at 260× *g* and RT for 5 min. For tissue samples, 100 mg of mice tissues or OCT-embedded human tissues from CRC patients were used. Cell or tissue samples were re-suspended in 500 µL of NZYol reagent by pipetting up and down, and incubated 5 min at RT. Subsequently, 100 µL of chloroform was added per sample, mixed by inversion, and incubated for 3 min at RT. Then, samples were centrifuged at 12,000× *g* and 4 °C for 15 min, and the upper phase, containing the isolated RNA, was transferred to a new tube and mixed with 1x volume of 70% ethanol. Then, RNA was purified with the RNeasy Mini Kit (Qiagen, Hilden, Germany), following the manufacturer’s instructions. RNA concentration was measured with the Nanodrop One (Thermo Fisher Scientific), and samples were stored at −80 °C until used.

cDNA synthesis was performed with 1 µg of total RNA using the NZY First-Strand cDNA Synthesis Kit (NZYtech, Lisbon, Portugal), following the manufacturer’s instructions. Oligo(dT)18 primers were used for the cDNA synthesis from CRC cells and mice tissues, whereas random hexamers were used for cDNA synthesis from human OCT-embedded tissues. Then, 0.8 µL of cDNA was used for semiquantitative PCR using 0.5 µM of the corresponding specific oligonucleotides (Appendix A) and 0.1 µL of the Phusion High-Fidelity DNA Polymerase (Thermo Fisher Scientific) in a final volume of 20 µL per reaction. For qPCR analyses, 2 µL of 1:10 dilution of cDNA were used for quantitation of mRNA levels of targeted genes, using 5 µL of TB Green Premix Ex Taq II (Takara) and 0.2 µL of each 10 µM oligonucleotide per reaction. PCR analyses were performed with the ProFlex PCR system (Applied Biosystems, Waltham, MA, USA), whereas Light Cycler 480 (Roche, Basel, Switzerland) was used for qPCR analyses. An 18S mRNA expression was used for normalization after the relative quantification of DNA band intensities with ImageJ software (version 1.52n).

### 2.7. In Vitro Functional Assays

Gain-of-function cell-based assays were performed with CRC cells stably overexpressing SPRYD7, with mock-stably transfected cells as control, as previously described [14,18,21]. Alternatively, loss-of-function cell-based assays were performed 24 h after transient transfection of CRC cells with SPRYD7 or control siRNAs [22]. Briefly, proliferation assays were performed in 96-well plates, seeding 1 × 10^4^ cells per well in 100 µL of complete DMEM. Plates were incubated at 37 °C and 5% CO_2_ for 72 h, and then 50 µL of 3 mg/mL MTT reagent ((3-(4,5-dimethylthiazol-2-yl)-2,5-diphenyl tetrazolium bromide, Sigma Aldrich) in complete DMEM was added per well. After 1 h incubation at 37 °C and 5% CO_2_, the medium was removed, and cells were lysed with 50 µL of DMSO (Merck) and incubated at RT at 100 rpm for 10 min. Finally, absorbance at 570 nm was recorded onto the Spark multimode microplate reader (Tecan Trading AG, Mannedorf, Switzerland).

Invasion assays were performed using 6.5 mm transwells with 8.0 µm Pore Polycarbonate Membrane Inserts (Corning, Corning, NY, USA) placed onto a 24-well plate coated with 35 µL of Matrigel matrix (Sigma-Aldrich) diluted in 1:3 in DMEM-free serum. A total of 1 × 10^6^ cells in 100 µL of sterile 0.5% BSA DMEM-free FBS (adhesion medium) were seeded onto pre-coated transwells in duplicate, and 700 µL of complete DMEM were added to each well as chemoattractant. After 22 h at 37 °C and 5% CO_2_, non-invading cells and Matrigel were removed, invasive cells on the lower membrane were fixed with 1 mL of 4% Paraformaldehyde (PFA) at RT for 30 min, and cells were stained with 1 mL of 0.2% crystal violet 25% methanol for 30 min at RT. Then, dye traces were removed, and invasive cells were counted under the DMi1 Microscope (Leica) with a 20x objective.

For adhesion assays, 100 µL 0.1 M NaHCO_3_, pH 8.8, containing Matrigel matrix (Sigma-Aldrich, 0.4 µg/mm^2^), were used to cover the wells of the 96-well plates. After O/N incubation at 4 °C and blocking with 200 µL of sterile 0.5% BSA DMEM FBS-free medium without phenol red (adhesion medium), 1 × 10^5^ cells in 50 µL of adhesion medium were seeded per pre-coated well and incubated for 2 h at 37 °C and 5% CO_2_. Cells were previously incubated in DMEM-free FBS for 24 h at 37 °C and 5% CO_2_, stained with 1 mg/mL BCECF (2′,7′-Bis-(2-Carboxyethyl)-5-(and-6)-Carboxyfluorescein, Acetoxymethyl Ester, Sigma-Aldrich) 1:100, diluted in DMEM FBS-free medium for 30 min at 37 °C and 5% CO_2_, and harvested with PBS 1x containing 4 mM EDTA. Then, detached cells were removed in subsequent washes with 100 µL of PBS 1x. Finally, attached cells were lysed with 50 µL of 1% SDS in PBS 1x, incubated 15 min at RT at 100 rpm, and cell fluorescence was read at 436 nm–535 nm excitation–emission, respectively, via the Spark multimode microplate reader (Tecan Trading AG).

Wound-healing assays for the analysis of the migratory capacity of CRC cells were performed with 2-well silicone inserts (Ibidi). A total of 2.5 × 10^5^ cells were placed in 70 µL of complete medium and seeded in each well of the inserts on 24-well plates. As many as 24 h later, inserts were removed, and 1 mL of complete medium was added per well. For the monitorization of cell migration, a Thunder imager (Leica, Wetzlar, Germany) with a constant 37 °C temperature and 5% CO_2_ concentration was used. Images of the wounds were taken every hour for 48 h. Each cell line was analyzed in duplicate. Finally, images were processed with the ImageJ program (Fiji) and the MRI wound healing tool. All calculated areas were visually inspected.

For colony soft agar assays, 6-well plates (Corning) were coated with 2 mL of sterile 0.5% noble agar (Condalab) in complete DMEM and incubated at RT for 15 min for agar solidification. Then, 1 mL of 0.5% noble agar solution was added to 2.5 × 10^4^ cells in 1 mL of complete DMEM, and 2 mL of cells were placed onto the solidified agar, in duplicate. After agar solidification at RT, cells were incubated at 37 °C and 5% CO_2_ for 4 weeks for colony formation, and 100 µL of complete DMEM medium supplemented with 0.6 mg/mL G418 were added to each well once a week. Finally, colonies were photographed with the DMi1 Microscope (Leica), and colonies were counted with the ImageJ program (Fiji).

Finally, tube formation assays were performed with HUVEC cells and the secretome of stably transfected SPRYD7 and mock CRC cells. The secretomes were collected after 48 h incubation at 37 °C and 5% CO_2_ of CRC cells in EGM-2 FBS-free medium supplemented with growth factors (Hydrocortisone, hFGF-B, VEGF, R3-IGF-1, Ascorbic acid, hEGF, GA-100 and Heparin). Then, secretomes (conditioned media) were centrifuged at 260× *g* for 5 min at RT and transferred to a new tube and stored at −20 °C until use. HUVEC cells were grown at 90% confluence in EGM-2 medium supplemented with 10% FBS; then, they were incubated for 24 h in complete EGM-2 FBS-free medium. After that, HUVEC cells were harvested with trypsin-EDTA and 2 × 10^4^ or 4 × 10^4^ cells were seeded in 96-well plates coated with 30 µL of Matrigel matrix (Sigma-Aldrich, 0.4 µg/mm^2^) per well for 1 h at 37 °C using EGM-2 FBS-free medium 1:2 diluted with the secretome of KM12 or SW cells. Plates were incubated at 37 °C and 0.5% CO_2_ for 6 h, and a DMi1 Microscope (Leica) was used to monitor tube formation each 2 h. Finally, images were processed with the ImageJ program (Fiji) and the angiogenesis analyzer tool. Each cell line was analyzed in duplicate.

### 2.8. 10-Plex TMT and LC-MS/MS

The 10-plex TMT (tandem mass tags) analyses of cells stably overexpressing SPRYD7 were performed with KM12C and KM12SM cells. KM12C mock-stably transfected and SPRYD7 overexpressing cells were labeled in triplicate, whereas KM12SM cells were labeled in duplicate. Cells were incubated in DMEM FBS-free medium for 48 h prior to cell collection and protein extraction.

TMT experiments and digestions were performed via the SP3 method as previously described [13,14] using 10 μg of each protein extract per TMT channel in 100 μL of RIPA. Briefly, after tris(2-carboxyethyl)phosphine (TCEP) reduction and alkylation with cloroacetamide, proteins were anchored to 100 µL of SeraMag magnetic beads mix (50% hydrophilic beads/50% hydrophobic beads, GE Healthcare), and O/N digested at 37 °C and 600 rpm with 0.5 μg of porcine trypsin (Thermo Fisher Scientific) in 100 μL of 200 mM HEPES, pH 8.0. Then, TMT labeling was performed separately (Thermo Fisher Scientific) in two incubation steps of 30 min at RT and 600 rpm with 10 μL of reagent per incubation, and a final incubation with 10 μL of 1 M glycine, pH 2.7, for 30 min at RT and 600 rpm was conducted. Finally, the contents of the 10 tubes were pooled together and dried under vacuum prior to peptide separation via High pH Reversed-Phase Peptide Fractionation Kit (Thermo Fisher Scientific) into 12 fractions of 300 µL each in a 0.1% triethylamine, 2.5–100% ACN gradient. Fractions were then mixed in six fractions by pooling the latest fractions with the initial ones (2, 9, and 1; 7 and 3; 10 and 4; 8 and 5; 11 and 6; 12), dried under vacuum, and stored at −80 °C until analysis in six LC-MS/MS runs using an Orbitrap Exploris 480 mass spectrometer (Thermo Fisher Scientific) equipped with the FAIMS (high field asymmetric waveform ion mobility spectrometry) Pro Duo interface technology.

Liquid chromatography (LC) peptide separation was performed prior to MS/MS analysis on the Vanquish Neo UHPLC System (Thermo Fisher Scientific). Samples were loaded into a precolumn PepMap 100 C18 3 µm, 75 µm × 2 cm Nanoviper Trap 1200BA (Thermo Fisher Scientific) and eluted in an Easy-Spray PepMap RSLC C18 2 µm, 75 µm × 50 cm (Thermo Fisher Scientific), heated at 50 °C. A total of 0.1% formic acid (FA) in H_2_O_mq_ was used as buffer A, and 0.1% FA in 80% can was used as as buffer B, whereas the mobile-phase flow rate was 300 nL/min. The 2 h elution gradient was 0–2% buffer B for 4 min; 2% buffer B for 2 min; 2–42% buffer B for 100 min; 42–72% buffer B for 14 min; 72–95% buffer B for 5 min; and 95% buffer B for 10 min. Samples were re-suspended in 10 µL of buffer A, and 3 µL of each sample were injected. For ionization, 2300 V of liquid junction voltage and 280 °C capillary temperature was used. The full-scan method employed a *m*/*z* 350–1400 mass selection, an Orbitrap resolution of 60,000 (at *m*/*z* 200), a AGC value of 300%, and maximum injection time (IT) of 25 ms. After the survey scan, the 12 most intense precursor ions were selected for MS/MS fragmentation. Fragmentation was performed with a normalized collision energy of 32, and MS/MS scans were acquired with 100 *m*/*z* first mass. The AGC target was 100%, the resolution was 15,000 (at *m*/*z* 200), the intensity threshold was 2 × 10^4^, the isolation window was 0.7 *m*/*z* units, and a maximum IT of 22 ms and the TurboTMT enabled. The charge state screening enabled us to reject unassigned, singly charged, and greater-than-or-equal-to-seven protonated ions. A dynamic exclusion time of 30 s was used to discriminate against previously selected ions. For FAIMS, a gas flow of 4.6 L/min and CV = −70 V and CV = −50 V were used.

### 2.9. Immunoprecipitation Coupled to LC-MS/MS

First, 50 µL of Protein G agarose beads (Santa Cruz Biotechnology) were washed 3 times with 10× beads volume of RIPA buffer and re-suspended in 1× beads volume of RIPA. Three mg of each protein extract were then incubated with 1:2 of equilibrated beads to remove unspecific proteins for 4 h at 4 °C (precleaning). After precleaning, supernatants were transferred to a new tube and incubated with the anti-SPRYD7 monoclonal antibody O/N at 4 °C in a relation of 3 µg of antibody per 10 mg of protein extract (Appendix A). The next day, samples were incubated with 1:2 of equilibrated beads for 3 h at 4 °C and in rotation and washed 4 times with 1x beads volume of RIPA, and interacting proteins were eluted twice in 50 µL of 50 mM Tris-HCl, pH 6.8, 10% glycerol, 0.1% SDS, 1.5% β-mercaptoethanol buffer after incubation at 95 °C for 5 min. In addition, proteins non-specifically attached to the beads used during the precleaning step were also eluted and used as negative controls of the IP. All incubations were performed in rotation, and all supernatants were discarded or recovered via centrifugation at 1000× *g* and 4 °C for 2 min.

Finally, for the identification of SPRYD7 interacting proteins, 50 µL of eluted proteins was analyzed by LC-MS/MS, whereas 10 µL was used for 10% SDS-PAGE and silver staining to confirm the presence of interacting proteins prior to LC-MS/MS. In addition, a second IP was performed with KM12C, KM12SM, SW480, and SW620 CRC cells stably overexpressing SPRYD7 for the validation of interacting proteins by WB.

Prior to LC-MS/MS, 50 µL of eluted proteins was trypsin digested via the SP3 method, as described above. After digestion, peptide samples were desalted using Pierce Peptide Desalting Spin Columns (Thermo Fisher Scientific) following the manufacturer’s instructions, dried under vacuum centrifugation, and re-suspended in 5 μL of 0.1% FA and 2% ACN prior to analysis via nano LC-MS/MS using the Q Exactive mass spectrometer (Thermo Fisher Scientific) of the Proteomics and Genomics Facility of the Center for Biological Research (CIB-CSIC) according to established protocols [24].

### 2.10. Mass Spectrometry Data Analysis

For the analysis of MS raw data, MaxQuant software (version 2.1.3) was used with standardized workflows [20]. Searches were performed against the Uniprot UP000005640_9606.fasta Homo sapiens (human) 2022 database (20,577 protein entries) using Reporter ion MS2 type for TMT and standard type for IP analyses.

TMT data were normalized via the sample loading normalization (SL) method to equal the differences in the total sum of signals of each TMT channel with the R Studio program (version 4.1.1) according to established protocols (https://github.com/pwilmart, accessed on 2 November 2022), using “tidyverse”, “psych”, “gridExtra”, “scales”, and “ggplot2” packages (Appendix A). In addition, principal component analysis (PCA) was performed to determine sample clustering using the “stats” R package prior to statistical analysis (Appendix A).

For statistical analysis of TMT data, moderated t-statistics analysis was performed with R Studio (version 4.1.1) using the packages “dplyr”, “tidyverse”, “limma”, “edgeR”, “ggplot2”, and “rstatix” after removing of reverse and contaminant proteins, data filtering (proteins identified in at least 50% of samples were considered for the analysis), and missing values imputation via random draws from a gaussian using the “imputeLCMD” R package. Proteins identified with one or more unique peptides, an expression ratio ≥ 1.5 or ≤0.67, and an adjusted *p*-value (FDR) ≤ 0.05 were selected as statistically significant dysregulated proteins. Expression ratio cut-offs were selected according to previous studies [13,21,25].

Regarding the IP mass spectrometry analysis, proteins identified in the negative control and among SPRYD7 interactors were discarded for the analysis. In addition, potential SPRYD7 interactors identified from the IP were first analyzed using the CRAPome database (contaminant repository for affinity purification MS data) to remove false positive interacting proteins from the analysis [26]. Proteins identified in >15% of the IP experiments from this database were discarded for further analysis.

### 2.11. Bioinformatics Analysis

Proteins identified as dysregulated, associated with SPRYD7 stable overexpression via TMT analysis, or as potential interactors were further investigated using STRING (version 11.0) and DAVID (version 6.8) databases. STRING [27] provides information about clusters of direct or indirect interaction in which dysregulated proteins are significantly involved, whereas DAVID [28,29] was used to identify altered networks and pathways more prone to be dysregulated. For the analysis with STRING, settings were fixed to 2 MCL clustering enrichment and 0.4 confidence score.

TMT expression ratios from each TMT experiment were represented as a heatmap using the MultiExperiment Viewer (MeV, version 4.9.0).

### 2.12. In Vivo Animal Experiments

Protocols used for the experimental work with mice were approved by The Ethical Committee of the Instituto de Salud Carlos III (Spain) after approval by the OEBA ethical committee (Proex 285/19).

Nude mice were used to analyze the role of SPRYD7 in CRC metastasis in vivo by means of liver homing and subcutaneous experiments according to established protocols [14,18]. Briefly, for in vivo liver homing, 1x10^6^ CRC cells stably overexpressing SPRYD7 and mock-stably transfected KM12C and KM12SM cells were intrasplenically injected in 100 µL of PBS 1x. Mice (n = 2 per condition) were euthanized with CO_2_ 24 h after inoculation of KM12C or KM12SM SPRYD7 or mock-stably transfected cells, the RNA from the liver and spleen was obtained, and the presence of cancer cells was analyzed via PCR using specific indicated oligonucleotides (Appendix A). Murine β-actin (mβActin) was used as the control of the assay and for normalization.

For the induction of subcutaneous tumor formation (n = 6), 1 × 10^6^ KM12C or KM12SM CRC cells stably overexpressing SPRYD7 and mock-stably transfected cells were subcutaneously injected, tumors were measured with an external caliper twice a week, and their volumes were calculated as (width)^2^ × (length). All mice were euthanized with CO_2_, and tumors were excised and paraffin-embedded for further evaluation when tumors reached 1500 mm^3^. In addition, the expressions of Ki67, SPRYD7, CD31, and CD34 in FFPE tumors were analyzed via immunohistochemistry (IHC) using specific antibodies (Appendix A).

### 2.13. Tissue Microarrays and Immunohistochemistry

Two tissue microarrays (TMA) containing (i) 55 cores from non-recurrence CRC and 28 cores from recurrence CRC; and (ii) 24 cores from non-metastatic CRC samples and 19 cores from liver metastasis tissue samples were used for IHC staining [30]. Slides were deparaffinized and incubated with the corresponding antibodies at optimized dilution (Appendix A), as previously described [24,30]. Sections were then visualized with 3,3′-diaminobenzidine for 5 min and counterstained with hematoxylin. Immunoreactivity was graded as 0, absent; 1, mild staining; 2, moderate staining; or 3, intense staining. Cases were classified according to total staining (intensity of the staining per percentage of areas showing reaction). In all cases, an external negative control was included, and slides were also incubated with the HRP-conjugated secondary antibody as negative control, which confirmed the absence of background signal. TMAs were scanned with the NanoZoomer scanner (Hamamatsu photonics), and 40× images were processed with the NDP.view 2 software (version 2.7.25).

### 2.14. Statistical Analysis

Microsoft Excel 2019 and GraphPad Prims 5 programs were used for statistical analysis (mean, standard error of the mean (SEM), and Student’s *t* test) and to obtain the plots of the IHC, alongside in vitro and in vivo functional assays. *p*-values ≤ 0.05 were considered statistically significant.

## 3. Results

### 3.1. SPRYD7 Dysregulation in Colorectal Cancer

SPRYD7 was described in a multi-dimensional proteomics analysis of CRC metastasis as significantly upregulated in the membrane compartment of KM12SM liver-metastatic CRC cells in comparison to isogenic poorly metastatic KM12C cells (Figure 1A) [15].

Here, we hypothesized that SPRYD7 would have a significant role in CRC progression and metastasis. To address this question, we confirmed the association of a high expression of SPRYD7 with CRC progression in other CRC cells and in actual tumoral samples of CRC patients. Then, we focused on the characterization of SPRYD7 and on the analysis of its functional role in vitro and in vivo in CRC progression and metastasis.

First, we analyzed a collection of different CRC cell lines at mRNA and protein level to confirm previous proteomics data and to determine whether the expression of SPRYD7 was observed beyond the KM12 cell system. Via qPCR, SPRYD7 was found to be upregulated at the mRNA level in highly metastatic-to-liver KM12SM and to liver and lung highly metastatic KM12L4a CRC cells in comparison to their isogenic poorly metastatic KM12C CRC cells. Moreover, higher mRNA levels of SPRYD7 were found in SW48 and Lim-1215 CRC cells derived from a stage-IV tumor and a metastatic site [31,32], respectively, in comparison to poorly metastatic HT-29, Caco-2, and RKO CRC cells. In addition, and in concordance with the KM12 cell model, the highly metastatic-to-lymph-nodes SW620 CRC cells showed higher SPRYD7 mRNA levels in comparison to their isogenic non-metastatic SW480 cell pair (Figure 1B). Moreover, besides SPRYD7 overexpression in KM12SM cells in comparison to KM12C poorly metastatic cells observed by proteomics (Figure 1A), SPRYD7 showed higher protein expression levels in the highly metastatic SW620 cells in comparison to its corresponding isogenic non-metastatic pair SW480 CRC cells (Figure 1C). Furthermore, SW48 and Lim-1215 CRC cells showed higher SPRYD7 protein levels in comparison to poorly metastatic Caco-2 and RKO cells, as previously observed at the mRNA level. In contrast to the results observed at the mRNA level, HT-29 cells showed the highest protein expression levels, suggesting a post-translational regulation of SPRYD7 in this cell line. These results suggest that the dysregulation of SPRYD7 is a common event in CRC, i.e., not restricted to the KM12 cell model of CRC metastasis, being mostly associated with the metastatic capacity of CRC cell lines.

Then, we focused on determining whether the dysregulation of SPRYD7 would have an impact on patients’ survival at the mRNA and/or protein level. At the mRNA level, from analysis of the GSE17538 cohort containing colon and rectum adenocarcinoma samples, high expression of SPRYD7 was significantly associated with the poor survival of CRC patients (*p*-value = 0.00016, Figure 1D). Next, to further elucidate its clinical relevance in CRC, two TMAs were analyzed. Although SPRYD7 was not associated with recurrence of CRC patients, SPRYD7 was found significantly highly expressed in CRC patients suffering from liver metastasis (Figure 1E). Additionally, a non-significant trend to poor survival of CRC patients was observed according to the high protein levels of SPRYD7 (*p*-value = 0.155) (Figure 1E), confirming previous findings at the mRNA level. Finally, higher mRNA levels of SPRYD7 were found in tumor tissue samples of CRC patients in comparison to paired non-tumoral tissues (Figure 1F).

Collectively, these results suggest a relevant association of SPRYD7 with CRC and CRC metastasis and aggressiveness.

### 3.2. Effect of Stably SPRYD7 Upregulation or Transient SPRYD7 Depletion on the Properties of CRC Cells

Then, to investigate the role of SPRYD7 in CRC and CRC metastasis, the stable overexpression of SPRYD7 was induced in the SW480/SW620 CRC cell pair and in the KM12C and KM12SM CRC cell lines as a model of CRC liver metastasis. First, mRNA and protein expression were analyzed via PCR, qPCR, and WB in SPRYD7-stably transfected CRC cells in comparison to mock-stably transfected control cells (Figure 2A). SPRYD7 upregulation was confirmed at the mRNA and protein level in the four CRC cell lines, highlighting a pronounced increase in SPRYD7 in the KM12C poorly metastatic and SW480 non-metastatic cells.

Then, the effect of SPRYD7 in the tumorigenic and metastatic properties of KM12 and SW CRC cell models was analyzed by means of proliferation, colony formation, migration, invasion, and adhesion cell-based assays. First, we observed that highly metastatic-to-liver KM12SM cells were more proliferative, migratory, invasive, and adherent than their isogenic poorly metastatic KM12C cells. In contrast, SW480 cells were more proliferative and migratory than their isogenic highly metastatic-to-lymphatic-nodes SW620 cells, whereas no differences were observed in the invasive capacity of these isogenic SW CRC cells. In addition, SW620 cells possessed a higher adhesion capacity than the poorly metastatic SW480 cells. Interestingly, both KM12SM and SW620 cells formed a lower number of colonies but these were larger than their corresponding poorly metastatic KM12C and non-metastatic SW480 cells.

Regarding the effect of SPRYD7 in the tumorigenic capacity of CRC cells, slight differences in the proliferation rate of SPRYD7-stably transfected cells in comparison to control cells were observed, whereas SPRYD7-stably transfected CRC cells significantly increased their anchorage-independent growth capacity by 1.3-fold in comparison to mock control cells (Figure 2B and Appendix A). Moreover, regarding their metastatic capacities, slight differences in the migration capacity of CRC cells were observed upon SPRYD7 overexpression, whereas an increased significant invasive capacity was observed after SPRYD7 overexpression (Figure 2B). Additionally, a significant increase (>1.5-fold) in the adhesion capacity of all CRC cells was observed upon SPRYD7 overexpression (Figure 2B and Appendix A).

Remarkably, the effects of SPRYD7 upregulation were more striking in poorly or non-metastatic KM12C and SW480 CRC cells, respectively, as these cells improved their tumorigenic and metastatic properties, matching or surpassing those of their corresponding isogenic metastatic KM12SM and SW620 CRC cells, respectively. Interestingly, SPRYD7 induced both adhesion and invasion. Adhesion might be associated with the late events of the metastatic colonization, where cells need to increase their adhesive properties to attached themselves to the organ of colonization [14,33,34,35], whereas invasive properties should be associated with a higher capacity of penetration in tissue for colonization [33,34,35].

Finally, SPRYD7 transient silencing via siRNAs was induced. First, depletion of SPRYD7 via transient silencing was efficiently achieved, as observed via qPCR, PCR, and WB analyses (Figure 2C). Then, tumorigenic and metastatic properties were assessed to further confirm the influence of SPRYD7 in CRC (Figure 2D). Since the most important effects on cells were related to invasion, adhesion, and wound healing, these assays were performed to confirm the involvement of SPRYD7 in these processes. An opposite significant behavior regarding previous findings upregulating SPRYD7 were observed. SPRYD7 depletion significantly altered the invasion, adhesion, and migration of CRC cells in comparison with siRNA control to similarly opposite extents to those observed with the stable overexpression of SPRYD7 (Figure 2D).

Collectively, in vitro results upon stable overexpression of SPRYD7 and transient SPRYD7 silencing support the previous hypothesis of a potential role of SPRYD7 in CRC development and metastasis.

### 3.3. In Vivo Analysis of SPRYD7 Role in Tumor Growth and Metastasis

Then, we performed in vivo assays to further address the role of SPRYD7 in CRC. To this end, we only focused on the KM12 cell model of liver metastasis to evaluate changes in liver homing and tumor growth [14,18,36]. First, liver homing analyses were performed to evaluate the capacity of stably transfected SPRYD7 cells to migrate to the liver in comparison to mock cells. KM12C and KM12SM mock cells were used as negative and positive controls of liver homing metastasis, respectively, and the detection of human GAPDH via PCR was used as a surrogate marker of the presence of human CRC cells in the liver. KM12C stably transfected SPRYD7 cells showed an increased capacity to migrate to the liver 24 h after the intrasplenic injection of cells (Figure 3A). In addition, both mock and stably overexpressing KM12SM cells were able to migrate to the liver at similar extents (Figure 3A).

Then, the subcutaneous inoculation of SPRYD7-stably transfected cells and mock cells was performed to evaluate in vivo alterations in the tumor growth associated with SPRYD7 overexpression. Importantly, higher measurable subcutaneous tumors were observed in all mice inoculated with SPRYD7-stably overexpressing cells than with mock control cells in terms of size, volume, and tumor weight (Figure 3B). In addition, SPRYD7-stably transfected KM12C cells almost equaled the tumor growth ability of KM12SM control cells. Remarkably, tumors induced via SPRYD7-stably transfected KM12C and KM12SM cells were visually more vascularized than mock-stably transfected cells induced tumors, suggesting a potential role for SPRYD7 in blood vessels formation.

Finally, subcutaneous tumors were analyzed via IHC. First, SPRYD7 overexpression in KM12C and KM12SM SPRYD7-stably transfected cells was confirmed in comparison to their control cells at endpoint (Figure 3C). In addition, higher levels of Ki67 were observed in SPRYD7-stably transfected KM12C and KM12SM tumors in comparison to mock control tumors, confirming the increased proliferation capacity of tumors derived from SPRYD7 overexpressing CRC cells (Figure 3C). Finally, to further investigate the visual increased vascularization of tumors induced via SPRYD7-stably transfected CRC cells, tumors were stained with two markers of vascular endothelial cells: CD31 and CD34. Both markers were found to be overexpressed in tumors induced via SPRYD7-stably transfected CRC cells, suggesting the potential of SPRYD7 to induce angiogenesis (Figure 3C). Interestingly, effects were more noticeable in KM12C cells overexpressing SPRYD7 than in KM12SM cells, as previously observed in the in vitro functional assays.

These results confirmed the involvement of SPRYD7 in CRC progression and CRC metastasis, and according to the results, a role was suggested for SPRYD7 in CRC cell migration, invasion, and tumor growth, with SPRYD7 favoring liver homing, as well as a potential role for SPRYD7 in the induction of angiogenesis.

### 3.4. Validation of SPRYD7 Role in CRC Associated Angiogenesis

As previous in vivo results suggested a role of SPRYD7 in angiogenesis, tube formation assays were performed with SPRYD7 and mock-stably transfected KM12 and SW cells to further confirm its potential role in angiogenesis.

The secretome of KM12C, KM12SM, and SW480 CRC cells stably overexpressing SPRYD7 increased the tube formation capacity of HUVEC cells in comparison with the secretome of mock cells, inducing a significantly larger number of branches, segments, meshes, junctions, and nodes than mock-stably transfected cells (Figure 4). In contrast, neither the secretome of mock-nor SPRYD7-stably overexpressing SW620 cells induced the tube formation capacity of HUVEC cells. The latter could be associated with the metastatic nature to lymph nodes of SW620 CRC cells. SW620 cells were isolated from a lymph node metastasis 1-year apart than SW480 cells isolated from the primary tumor [37,38]. Thus, SW620 CRC cells might not need to induce angiogenesis and form more tubes to proliferate, migrate, and colonize other organs than SW480 cells.

Additionally, the secretome of SW480, SW620, KM12C, and KM12SM cells upon transient SPRYD7 silencing via siRNAs was also analyzed for their tube-formation ability (Appendix A). As also observed in the invasion, wound healing, and adhesion assays, the transient depletion of SPRYD7 reduced the ability of HUVEC cells to form tubes. Although the results were non-significant, an opposite trend to the results obtained via overexpressing SPRYD7 was observed for SW480, KM12C, and KM12SM CRC cells (Appendix A). Again, the secretome of SW620 cells upon depletion of SPRYD7 or control was almost unable to induce the tube-formation capacity of HUVEC cells.

Collectively, these results, together with the in vivo assays and IHC, confirmed the role of SPRYD7 in angiogenesis. The stable overexpression of SPRYD7 was able to induce angiogenesis, especially in the poorly metastatic cells, whereas the transient depletion of SPRYD7 reduced the angiogenesis capacity of CRC cells. Therefore, SPRYD7 effects on angiogenesis should allow for and facilitate cell migration and metastasis, which is in concordance with the results obtained in vivo.

### 3.5. Protein Dysregulation Profiling Promoted by SPRYD7 Overexpression in CRC Cells

As SPRYD7 is a barely studied protein, we next focused on determining its associated proteome to shed light on its role in CRC.

To this end, the proteome of SPRYD7-stably transfected KM12C and KM12SM cells was analyzed in comparison to mock-stably transfected cells to elucidate the proteins and biological pathways, dysregulated by SPRYD7 overexpression, which might be involved in CRC pathogenesis. Three biological replicates of stably overexpressing and mock KM12C cells and two biological replicates of KM12SM cells were analyzed via 10-plex TMT quantitative proteomics experiments. After trypsin digestion, peptide fractionation, and LC-MS/MS using an Orbitrap Exploris 480 equipped with FAIMS Pro Duo Interface, data were analyzed with MaxQuant, and statistically dysregulated proteins were identified using the R program. Importantly, despite the fact that confirmed via qPCR and WB the overexpression of SPRYD7 in KM12C and KM12SM CRC cells prior to proteomics analyses via TMT, we could not observe a significant overexpression of SPRYD7 (Appendix A) via mass spectrometry (1.01 ratio SPRYD7/mock in KM12C cells and 1.11 in KM12SM cells). This could be associated with the known phenomenon referred to as ratio compression, wherein distinct TMT-labeled peptide ions with close precursor *m*/*z* values could be co-isolated and co-fragmented in MS2, skewing reporter ion intensities towards a 1:1 ratio [39,40]. However, the PCA obtained prior to data analysis confirmed that the replicates clustered together, and a clear separation among stably overexpressing and mock control KM12C and KM12SM cells was observed (Appendix A), thus suggesting that the SPRYD7 TMT ratios would be masked by ratio compression, which might be behind the discrepancy observed between the proteomics data and those of qPCR and WB regarding SPRYD7 expression.

In total, 92 unique proteins out of the 5586 proteins identified and quantified with at least one unique peptide in more than 30% of the samples (Appendix A) were dysregulated in SPRYD7-stably transfected KM12 cells, with a SPRYD7/mock expression ratio of ≥1.5 and an FDR of <0.05 (Figure 5A). Although we observed proteins showing an opposite trend in their expression ratio between KM12C and KM12SM CRC cells, 53 of the dysregulated proteins in one cell line showed the same dysregulation in the other cell line but with a different expression ratio and FDR value (Figure 5B). The opposite trend would be associated with the different metastatic properties of the cells and the different protein levels of SPRYD7, which induced higher in vitro and in vivo effects on the poorly metastatic KM12C cells than on highly metastatic-to-liver KM12SM CRC cells. STRING revealed more than six clusters of significant direct or indirect protein interactions (FDR < 0.05), associated with cell cycle, oxidation–reduction processes, RNA processing, DNA replication, regulation of cytoskeleton, and anchoring junctions (Appendix A). Additionally, gene ontology analysis showed that these proteins were involved in transcription, DNA replication, immune response, or protein localization (Appendix A), i.e., processes related to the pathogenesis of cancer cells with mainly nuclear and exosome proteins. Subsequently, four dysregulated proteins were selected for validation based on their expression ratio, biological process, and their previous association to CRC or other cancers (Table 1 and Appendix A). WB analyses with protein extracts from mock and SPRYD7-stably overexpressing transfected KM12C, KM12SM, SW480, and SW620 CRC cells confirmed the upregulation of KDELC2, SETD8, and DDB2, and the downregulation of IRF9 in CRC cells overexpressing SPRYD7, thus confirming the modulation of the protein expression levels of the analyzed proteins via SPRYD7 overexpression (Figure 5C). However, in some cases, this dysregulation was more noticeable in the poorly metastatic KM12C and SW480 cells and in the highly metastatic-to-liver KM12SM cells than for the high metastatic to lymph nodes SW620 cells.

### 3.6. Interactome Associated with SPRYD7

To further characterize the interaction network of SPRYD7 and investigate the processes in which it might be involved, we finally carried out an immunoproteomics approach. To this end, SPRYD7-stably transfected KM12C and KM12SM CRC cells were used. Prior to the IP, protein extracts were analyzed via Coomassie blue staining and WB to confirm their quality for the IP assay (Appendix A). In addition, prior to LC-MS/MS analysis, proteins eluted from the IPs were visualized via silver staining after SDS-PAGE to confirm the presence of potential interacting proteins and differential protein profiles in comparison to the negative controls of the assay (Appendix A). After LC-MS/MS analysis, SPRYD7 was observed among the identified interacting proteins, highlighting the quality of the assays.

In total, the analysis of the interactome associated with SPRYD7 allowed us to identify 118 and 157 proteins as SPRYD7 interactors from KM12C and KM12SM protein extracts, respectively. Interestingly, after CRAPome data analysis, 133 proteins were selected as potential SPRYD7 interactors, with 8 of them identified in both IP analyses (Table 2 and Appendix A). Using the STRING database and data mining, six significant clusters of direct or indirect protein interactions were identified (FDR < 0.05), related to actin filament, IL-1 signaling, exocytosis, muscle contraction, metabolic and biosynthetic processes, and focal adhesions (Figure 6A). Remarkably, none of these proteins have been previously described as SPRYD7 interactors. Subsequent enrichment analysis via the DAVID database revealed proteolysis, immune response, cell adhesion, or actin cytoskeleton as the biological processes closely associated with CRC pathogenesis related to the overexpression of SPRYD7 (Appendix A).

Finally, 11 potential interactors were selected for validation according to their biological role, previous association to CRC or other cancers, and antibody availability. WB analyses confirmed the eleven candidate proteins as SPRYD7 interactors from KM12C, KM12SM, SW480, and SW620 CRC cells stably overexpressing SPRYD7 immunoprecipitations (Figure 6B). In addition, two cross-validation IPs performed with the ILR1N and ARG1 antibodies further confirmed these results, suggesting that validated candidate proteins are actual SPRYD7 interactors (Figure 6C). In summary, WB confirmed proteomics results derived from the immunoprecipitation and from the TMT analysis, revealing a complex network of proteins interacting with, or modulated by, SPRYD7.

Collectively, the results presented here demonstrate a role of SPRYD7 in CRC progression and metastasis, highlighting its role as an inductor of angiogenesis, which makes this protein an interesting protein to be explored as a therapeutic target for the treatment of CRC patients.

## 4. Discussion

CRC is the third most common cancer and the second cause of cancer-associated death worldwide. Indeed, most CRC patients (about 60%) are still diagnosed at advanced stages (stages III and IV); the 5-years survival rate is lower than 50–10%, due mainly to liver metastasis; and chemotherapy and radiotherapy treatments are needed. Thus, a deeper understanding of the molecular processes and changes occurring at late stages of the disease, which might lead to cancer metastasis, is necessary in order to identify new therapeutic targets or effective treatments of CRC. For such a purpose, the analyses of the proteome associated with cancer cells, tissue, or plasma samples via different proteomics techniques have been very useful for the identification of dysregulated proteins during CRC development [41,42,43,44].

Here, we have investigated the role in CRC progression and metastasis of a barely known protein, SPRYD7, not previously associated with CRC, which was described as upregulated by SILAC in the membrane subcellular compartments of highly metastatic KM12SM CRC cells in comparison to their isogenic poorly metastatic KM12C cells [15]. In this SILAC study, a spatial proteomics analysis was performed to address the changes in the abundance and localization of the proteome of the KM12 CRC cell system of liver metastasis [15]. SPRYD7 is a vesicular protein with only a SPRY domain and with an important role in regulating the immune system, which makes this protein an interesting target of study in CRC [19,45]. Therefore, we have shed light on the relevant association of SPRYD7 with CRC. In this sense, the high protein expression levels of SPRYD7 have been found associated with the high metastatic capacity of CRC cells and to poor patient prognosis, highlighting the importance of studying this dark protein to obtain a deeper understanding of pathological mechanisms associated with CRC progression and metastasis.

In vitro and in vivo assays evidenced an important role of SPRYD7 in CRC progression and metastasis. CRC cells overexpressing SPRYD7 were more adherent, possessing a higher anchorage-independent growth and invasive and migrative abilities than mock cells, equaling, in most cases, the highly metastatic KM12SM or SW620 cells. Moreover, transient siRNA SPRYD7 depletion followed by in vitro functional cell-based assays confirmed the involvement of SPRYD7 in the tumorigenic and metastatic properties of CRC cells. In addition, using the well-established KM12 cell model of CRC liver metastasis [36,38,46,47], an important impact on the metastatic capacity of SPRYD7-stably transfected CRC cells was observed in vivo. In this sense, the overexpression of SPRYD7 was able to promote the liver homing of poorly metastatic KM12C cells to a great extent. Remarkably, another goal of the study was to demonstrate that although the in vitro proliferation capacity of genetic engineered cells was not different from mock control cells, KM12C cells overexpressing SPRYD7 showed an increased tumor growth ability in vivo, equal to the growth capacity of KM12SM cells, which suggest that the overexpression of SPRYD7 induce changes in the tumoral microenvironment (TME), enhancing CRC development and progression. In this sense, a higher angiogenic capacity of SPRYD7-overexpressing cells was elucidated in vitro, using HUVEC cells, and in vivo, where subcutaneous tumors from SPRYD7-overexpressing cells were more vascularized and possessed increased levels of CD31 and CD34 angiogenic markers than tumors induced by mock control cells [48,49,50,51]. Furthermore, based on our results, SPRYD7 was observed to favor liver homing and tumor growth in vivo and to induce cell migration and invasion, which might also be associated with the high blood-vessel-formation capacity of cells overexpressing SPRYD7. This was confirmed by the higher induced angiogenic capacity of SPRYD7-stably transfected cells, which might promote cell proliferation and migration. Additionally, the transient depletion of SPRYD7 further confirmed the involvement of SPRYD7 in angiogenesis, invasion, migration, and adhesion, confirming an important role for SPRYD7 in CRC progression and metastasis.

Interestingly, the analysis of the differential proteome associated with SPRYD7 overexpression revealed several dysregulated biological processes in these cell lines in comparison to mock control cells, which might be directly or indirectly regulated upon SPRYD7 overexpression. The dysregulated pathways associated with SPRYD7 were related to the cell cycle, chromosome organization, RNA processing, regulation of cytoskeleton, or anchoring junctions. As SPRYD7 is a vesicle protein, it might have a potential role in cell communication; thus, it might be involved in the regulation of the above-indicated biological processes, identified here via TMT quantitative proteomics.

Analysis of the interactome associated with SPRYD7 was also performed to identify proteins that might be involved in the regulation of the biological processes identified as altered by SPRYD7 overexpression, which might be useful as targets of therapeutic intervention. In these analyses, the protein extracts from SPRYD7-stably transfected KM12C and KM12SM cells were used as these samples might be rich in SPRYD7 interacting proteins, and proteins non-specifically bound to the beads during the precleaning step were used as negative controls and removed from the analysis. The SPRYD7 interactors identified via LC-MS/MS were involved in immune and inflammatory responses, which might be related to the greater induced angiogenic capacity of CRC cells overexpressing SPRYD7, as well as in anchoring functions or actin cytoskeleton, as previously observed for proteins dysregulated due to SPRYD7 overexpression, suggesting the high importance of these pathways in the study of SPRYD7-associated roles in CRC. Interestingly, the proteomics identification of anchoring proteins associated with SPRYD7 upregulation is in concordance with the increased invasion and adhesion capacities of SPRYD7-stably overexpressing cells previously observed in the in vitro functional assays, which suggests that SPRYD7 might be involved in the changes in the cell adhesion patterns that occur during cell invasion [35,51,52,53]. In addition, only 60 of the 132 potential SPRYD7 interactors identified via mass spectrometry were also identified in the TMT quantitative proteomic analysis, the protein expression of five of them being dysregulated by SPRYD7 overexpression (i.e., SOD1, FSCN1, and P0DP25 upregulated and A2ML1 and SERPINB5 downregulated). Thus, SOD1, FSCN1, and A2ML1 might have a potential role in the SPRYD7-mediated role in CRC.

After confirmation via WB of the dysregulation by SPRYD7 of the indicated proteins and/or proteins interacting with SPRYD7, our results suggest a novel and interesting role of SPRYD7 in CRC progression and metastasis. SPRYD7 was here observed to be involved in invasion, adhesion, and migration in vitro and in liver homing and tumor growth in vivo, which might be enhanced by the high angiogenic capacity induced by SPRYD7 overexpression, as well as by the proteins identified here as dysregulated by SPRYD7 or as SPRYD7 interactors, which have also been demonstrated to be involved in inflammation or cancer.

In this sense, several of the proteins modulated or interacting with SPRYD7 described here have previously been associated with angiogenesis, inflammation, and cancer. Regarding angiogenesis, TYMP and ARG1 have been demonstrated to enhance angiogenesis by directly stimulating the expression of pro-angiogenic factors [54], or else by inducing angiogenesis and lymphangiogenesis through myeloid-derived suppressor cells [55]. Moreover, KDELC2 has also been demonstrated to upregulate glioblastoma angiogenesis via reactive oxygen species activation [56]. Furthermore, TYMP1 has also been described to possess an anti-apoptotic effect [54,57], whereas IL1RN has been described as an anti-inflammatory antagonist of the interleukin-1 family of proinflammatory cytokines protecting against immune dysregulation and uncontrolled systemic inflammation [58]. In cancer, TOLLIP has been described to promote hepatocellular carcinoma via the PI3K/AKT pathway [59], and NPHP3 has been shown to regulate cancer cell viability by mediating the primary cilium formation [60]. Moreover, SETD8 expression is enhanced in different carcinomas such as lung, renal, or gastric cancers [61]. Here, we observed via proteomics that SPRYD7 induced the expression of SETD8, which was further demonstrated via WB. Collectively, these data also support that SPRYD7 upregulation in CRC metastasis provokes the dysregulation of proteins associated with angiogenesis, inflammation, and cancer, contributing to the progression and aggressiveness of the observed phenotypes in CRC cells in vitro and in vivo, with enhanced liver homing and tumor growth.

Finally, as potential limitations of the study, although the role of SPRYD7 in CRC has been confirmed in this work via different in vitro and in vivo assays, further investigation of its role in the induction of angiogenesis and CRC progression might be performed via loss-of-function assays after a stable depletion of SPRYD7 in CRC cells to further confirm the here-obtained results. In addition, functional analyses of the role of the novel interactome and proteome associated with SPRYD7 overexpression in CRC should also be performed to further confirm their association with the SPRYD7-mediated process in CRC progression and metastasis.

## 5. Conclusions

We show here that SPRYD7 plays an important role in CRC and in CRC progression; thus, our results should increase interest in the role of SPRYD7 in CRC, as well as those of its interactors as potential therapeutic targets for CRC. Thus, the power and competence of these proteins and interesting interactors as therapeutic targets should be evaluated. Considering the data shown in this work, and upon confirmation of a role for SPRYD7 in angiogenesis induction, SPRYD7 might become an interesting target of intervention in blocking the angiogenic capacity of cancer cells, which might reduce their tumor growth and metastatic potential, with the aim of improving the prognosis and survival of CRC patients, as previously suggested for therapies based on antiangiogenic factors [62,63,64]. In this sense, future research exploring the feasibility of using pharmacological inhibitors and knockdown of SPRYD7 to assess the impact of SPRYD7 inhibition on cellular phenotypes, as well as to investigate the potential synergistic effect of targeting SPRYD7 in combination with standard CRC therapies, are warranted in order to explore the potential of SPRYD7 as a therapeutic target for CRC and for CRC metastasis treatment.

## Figures and Tables

**Figure 1 cells-12-02548-f001:**
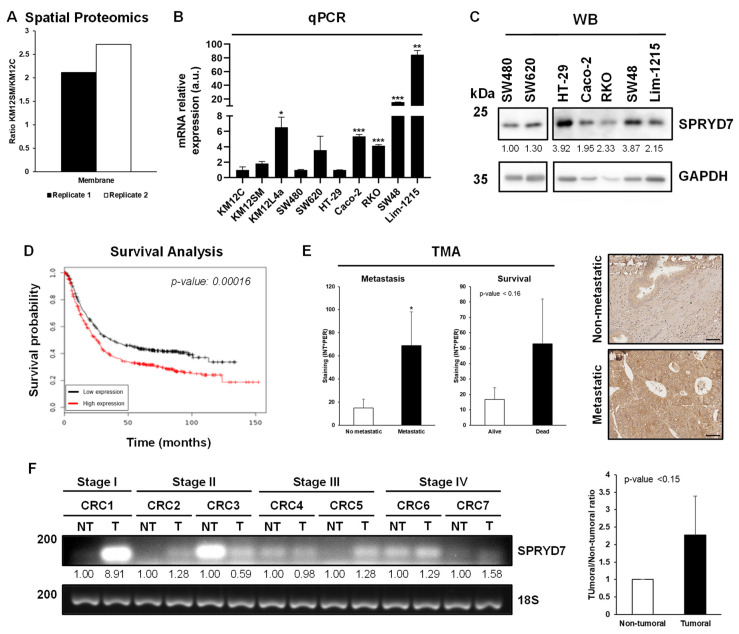
Association of SPRYD7 to colorectal cancer. (**A**) Higher protein expression levels of SPRYD7 were found in the membrane subcellular compartment by SILAC spatial proteomics analysis of KM12SM (highly metastatic to liver) cells in comparison to their isogenic KM12C (poorly metastatic) CRC cells in the forward (Replicate 1) and reverse (Replicate 2) experiments. mRNA (**B**) and protein (**C**) expression levels of SPRYD7 in CRC cell lines with different aggressiveness and metastatic properties. Higher mRNA and protein levels of SPRYD7 were associated with the higher aggressiveness and metastatic ability of CRC cells. (**D**) Data from the TCGA revealed a statistically significant association of SPRYD7 with a poor prognosis of CRC patients. (**E**) TMA analysis of CRC patients revealed an association of higher protein levels of SPRYD7 with liver metastasis and a poor survival of CRC patients. A representative image of non-metastatic and metastatic core is depicted. Scale bar: 50 µm. (**F**) PCR analysis of mRNA levels of SPRYD7 in paired non-tumoral (NT) and tumoral (T) tissues from CRC patients at different stages of the disease. *, *p*-value < 0.05; **, *p*-value < 0.01; ***, *p*-value < 0.001.

**Figure 2 cells-12-02548-f002:**
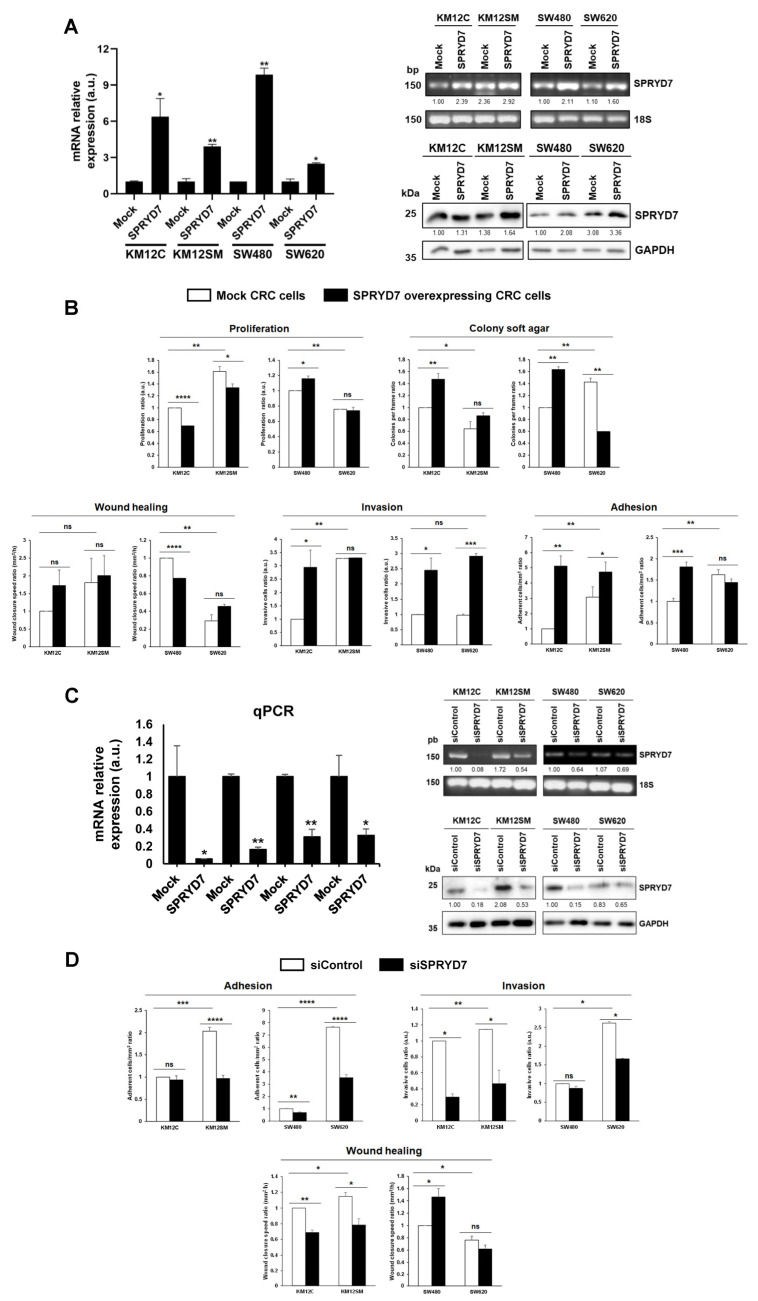
In vitro analyses of the role of SPRYD7 in CRC progression and metastasis. The overexpression of SPRYD7 (**A**) in the isogenic KM12 and SW CRC cell lines used in the study was confirmed via qPCR, PCR, and WB. (**B**) Gain-of-function assays revealed enhanced tumorigenic and metastatic properties of CRC cells upon SPRYD7 overexpression, despite meagre differences with respect to the proliferative and migratory capacity of these cells being observed. (**C**) Transient silencing of SPRYD7 achieved in the isogenic KM12 and SW CRC cells was confirmed via qPCR, PCR, and WB. (**D**) Loss-of-function invasion, adhesion, and wound healing assays revealed a decrease in the tumorigenic and metastatic properties of CRC cells upon transient silencing of SPRYD7. *, *p*-value < 0.05; **, *p*-value < 0.01; ***, *p*-value < 0.001; ****, *p*-value < 0.0001; ns, non-significant.

**Figure 3 cells-12-02548-f003:**
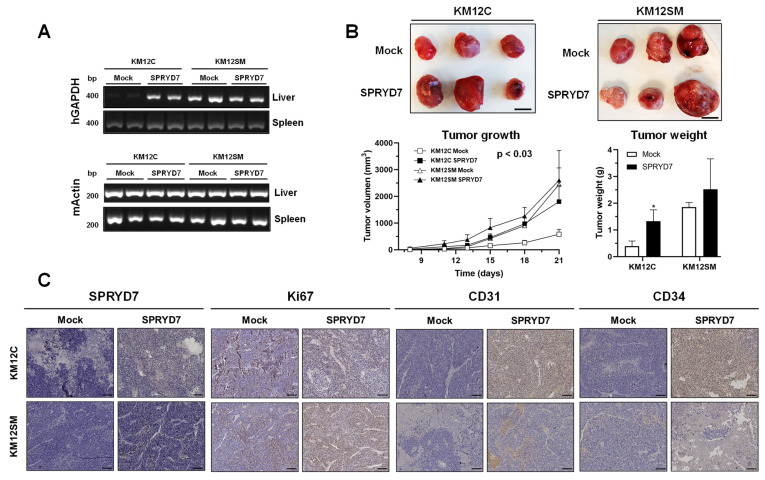
In vivo analysis of the role of SPRYD7 in tumor growth and metastasis. (**A**) PCR analysis with human GAPDH-specific oligonucleotides revealed the presence of human cells in the liver of SPRYD7-stably transfected KM12C and KM12SM cells 24 h after intrasplenic inoculation (n = 2 per group), in contrast to mock KM12C cells, suggesting a potential role of SPRYD7 in migration and liver homing. Mock KM12SM cells were used as positive control of homing assays, and murine actin specific oligonucleotides were used as loading control. (**B**) Representative images of three out of six subcutaneous tumors induced by mock- and SPRYD7-stably transfected cells. The higher size, volume, and weight of subcutaneous tumors grown after subcutaneous inoculation of SPRYD7-stably transfected cells in comparison to mock cells (n = 6 per group) confirmed the role of SPRYD proteins in CRC tumor growth in vivo. Scale bar: 10 mm. (**C**) Immunohistochemistry of paraffined subcutaneous tumors confirmed the overexpression of SPRYD7 in SPRYD7-stably transfected cells, whereas the overexpression of Ki67, CD31, and CD34 in tumors induced by these cells demonstrated the role of SPRYD7 in tumor proliferation and in the induction of angiogenesis. Scale bar: 100 µm. *, *p*-value < 0.05.

**Figure 4 cells-12-02548-f004:**
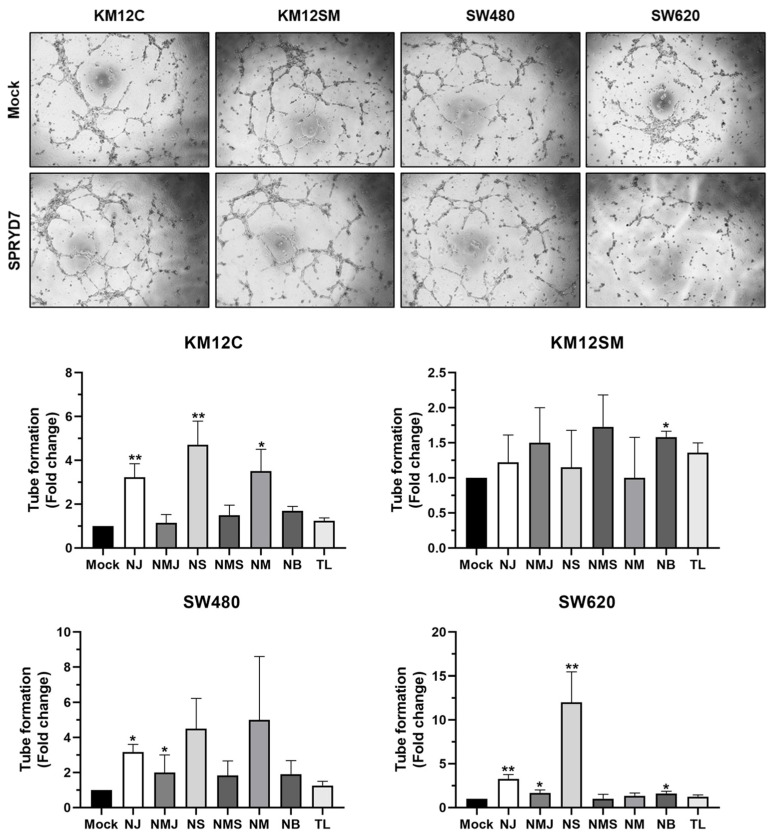
Validation of the role of SPRYD7 in angiogenesis. Tube formation assays confirmed the role of SPRYD7 in promoting angiogenesis, as an increased number of junctions, segments, meshed, and branches were observed for KM12C, KM12SM, SW480, and SW620 cells overexpressing SPRYD7. NJ, number of junctions; NMJ, number of master junctions; NS, number of segments; NMS, number of master segments; NM, number of meshed; NB, number of branches; TL, total length. *, *p*-value < 0.05; **, *p*-value < 0.01.

**Figure 5 cells-12-02548-f005:**
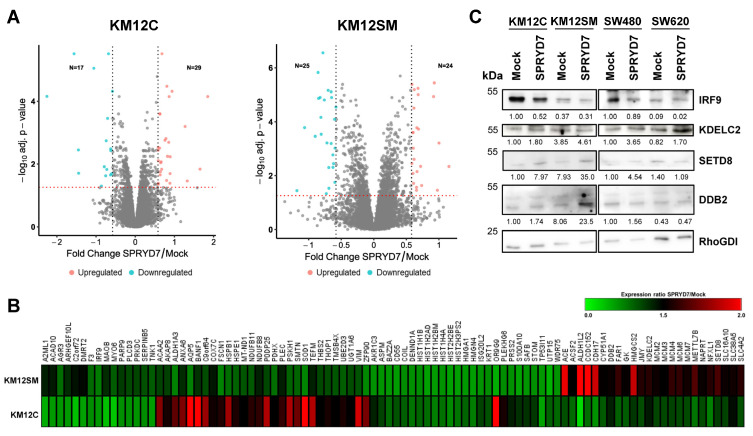
The 10-plex TMT quantitative proteomics analyses for the identification of dysregulated proteins induced by SPRYD7 upregulation. (**A**) Volcano plots of the protein profiles of KM12C and KM12SM CRC cells overexpressing SPRYD7 in comparison to mock control cells. The *x*-axis is log_2_ expression ratio (Fold change, FC), the *y*-axis is adjusted *p*-value (FDR) based on −log_10_. Proteins identified as upregulated (red) or downregulated (blue) in these cells with a ≥1.5-fold expression ratio (represented by two black dashed vertical lines) and an FDR < 0.05 (represented by a red dashed horizontal line) are shown. Grey dots represent proteins that do not fulfill the criteria for being classified as significantly dysregulated proteins. (**B**) Heatmap of the expression ratio of the 92 proteins identified as dysregulated by SPRYD7 overexpression in KM12C and/or KM12SM stably transfected cells. Upregulated and downregulated proteins are shown in red and green, respectively. (**C**) Western blot analyses of indicated proteins were performed with protein extracts from SPRYD7-stably overexpressing and mock CRC cells to validate the dysregulation of selected proteins induced by SPRYD7.

**Figure 6 cells-12-02548-f006:**
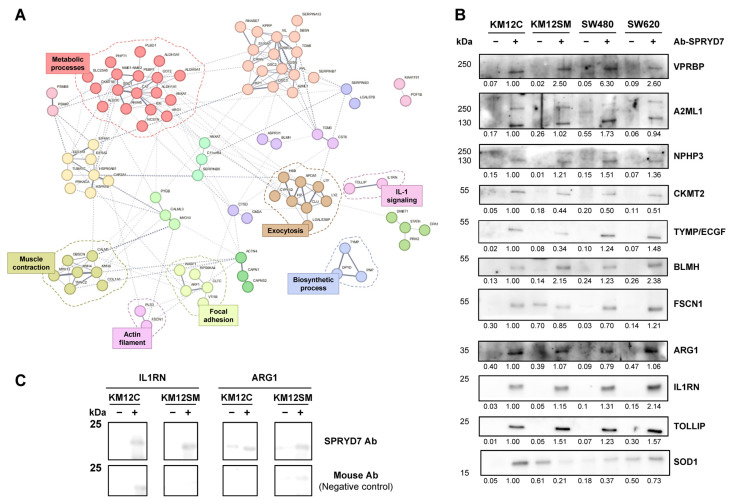
Analysis and validation of the interactome associated with SPRYD7. (**A**) STRING analysis for the elucidation of direct or indirect protein interactions previously described among SPRYD7 interacting proteins. Different clusters of interactions were defined for each protein, related to biological processes closely associated with CRC pathogenesis. (**B**) WB confirmation of the interaction between SPRYD7 and candidate proteins in the KM12C, KM12SM, SW480, and SW620 CRC cells as observed via immunoprecipitation with the anti-SPRYD7 antibody followed by WB analyses with antibodies against indicated proteins. (**C**) Cross-validation of the interaction of SPRYD7 with IL1RN and ARG1. The IP with antibodies specific to IL1RN or ARG1, followed by the detection of SPRYD7 with the anti-SPRYD7 antibody, allowed us to detect SPRYD7 among their interacting proteins, thus further validating IL1RN and ARG1 as SPRYD7 interactors.

**Table 1 cells-12-02548-t001:** List of dysregulated proteins selected for validation from TMT analysis.

Gene	Protein	KM12C SPRYD7/Mock	KM12SM SPRYD7/Mock
**IRF9**	Q00978	0.60	0.77
**DDB2**	Q92466	1.31	1.52
**KDELC2**	Q7Z4H8	1.36	1.58
**SETD8**	Q9NQR1	1.14	1.65

**Table 2 cells-12-02548-t002:** List of potential SPRYD7 interacting proteins selected for validation from the IP assay.

Protein	Gene	Cell Line
**Q7Z494**	NPHP3	KM12SM
**Q5W111 ***	SPRYD7	KM12SM
**P19971**	TYMP	KM12C
**A8K2U0**	A2ML1	KM12C, KM12SM
**Q9H0E2**	TOLLIP	KM12SM
**P05089**	ARG1	KM12C
**Q13867**	BLMH	KM12C
**P00441**	SOD1	KM12SM
**Q16658**	FSCN1	KM12C
**Q9Y4B6**	VPRBP	KM12SM
**Q96L46**	CAPNS2	KM12SM
**P18510**	IL1RN	KM12SM

* SPRYD7 was identified among the proteins in the IP, validating the quality of the assay.

## Data Availability

The data presented in this study are available in the main body of the manuscript and in the Appendix A. Proteomics data are deposited at the ProteomeXchange Consortium repository via the PRIDE under the datasets PXD042874 for the IP and PXD042901 for the TMT analyses [65].

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
