# Peer review of "Functional Proteomics Characterization of the Role of SPRYD7 in Colorectal Cancer Progression and Metastasis"

_cells, 2023, doi:10.3390/cells12212548_

Round 1
Reviewer 1 Report
The authors studied the role of SPRYD7 in colorectal cancer progression and metastasis by using proteomics techniques; however, the data is still not sufficient to address the conclusion and the writing need to be improved. The major concerns are shown below.
1. In figure 1, the author compared the expression of SPRYD7 in different cell lines, but the author did not show the protein expression in KM12C and Km12SM cells. Why SPRYD7 is mostly expressed in HT29 cells that has low aggressive and low metastatic properties? How the author defined the metastatic activity, the author may detect the transwell or wound-healing activity and then examine the correction between the metastatic capacity and SPRYD7 expression.
2. The author examined the adhesion activity of CRC cells; however, high adhesion activity may inhibit the invasion capacity, which is not consistent with the conclusion. The author should show the images of Figure 2B in the manuscript or supplementary material.
3. In Figure 3, the author use 2 mice for Figure 3A and 3 mice for Figure 3B? The animal studies usually use 5 mice per group for statistical analyses.
4. The employed proteomics analyses in Figure 5 and Figure 6; however, there are no relationship between these results, and there are no functional validation of these proteins, such as CRC proliferation and migration assays.
5. The author only performed SPRYD7 overexpression experiments in these CRC cells, the author should also design knockdown experiments in the cells that has a high expression level of SPRYD7.
Author Response
Reviewer 1
Comments and Suggestions for Authors
The authors studied the role of SPRYD7 in colorectal cancer progression and metastasis by using proteomics techniques; however, the data is still not sufficient to address the conclusion and the writing need to be improved. The major concerns are shown below.
We are very grateful to the reviewer for his/her work, for his/her comments, and for the useful suggestions pointed out to improve the manuscript.
- In figure 1, the author compared the expression of SPRYD7 in different cell lines, but the author did not show the protein expression in KM12C and Km12SM cells. Why SPRYD7 is mostly expressed in HT29 cells that haslow aggressive and low metastatic properties? How the author defined the metastatic activity, the author may detect the transwell or wound-healing activity and then examine the correction between the metastatic capacity and SPRYD7 expression.
SPRYD7 was found upregulated in the highly metastatic to liver KM12SM cells in comparison to their isogenic poorly metastatic KM12C cells in a previous quantitative proteomics analysis, as depicted in Figure 1A (Mendes, Pelaez-Garcia et al. 2017). Therefore, we included the analysis of the protein levels of SPRYD7 by proteomics in the manuscript. Then, we performed the analysis of the SPRYD7 mRNA levels by qPCR in ten colorectal cancer cell lines including the KM12 cell model. Additionally, in the western blot analysis performed to confirm the overexpression of SPRYD7 in the four CRC cell lines after transfection (Figure 2A), the upregulation of SPRYD7 in the Mock KM12SM cells in comparison to their isogenic poorly metastatic Mock KM12C CRC cells can also be observed, confirming the previous proteomics data. Al these results confirmed the upregulation of SPRYD7 in KM12SM highly metastatic to liver CRC cells in comparison to KM12C non-metastatic cells.
Regarding SPRYD7 protein expression in HT-29 cells, we suggested that the SPRYD7 expression in the HT-29 cell line at protein level might be regulated by a post-translational modification mechanism because lower mRNA levels of SPRYD7 were observed in this cell line in comparison to the other CRC cells. As such, we have highlighted this information in the revised version of the manuscript. Moreover, we analyzed SPRYD7 levels in 10 CRC cells that were classified according to their metastatic properties regarding previous scientific literature. Therefore, to address the concern of the reviewer, we have included two novel references in the revised version of the manuscript highlighting that SW48 and LIM1215 cells derived from a stage IV tumor and a metastatic site, respectively.
Finally, for the in vitro functional assays, both Mock and SPRYD7-stably overexpressing cells were used, and the effect of SPRYD7 in the tumorigenic and metastatic capacities of KM12C, KM12SM, SW480, and SW620 cells was obtained by normalizing the proliferation, adhesion, colony formation, migration, and invasion capacity of cells overexpressing SPRYD7 to their corresponding Mock cells. However, to clarify, and as required by Reviewer 2, we have modified the Figure 2B to include the normalization of the proliferation, adhesion, colony formation, migration, and invasion capacity of all CRC cells to the poorly-metastatic Mock KM12C or the non-metastatic SW480 cells, which allows for a better comparison between Mock isogenic cells, and for a better understanding of the effect of SPRYD7 overexpression in the tumorigenic and metastatic properties of KM12 and SW CRC cells.
To address the concern of the reviewer, we have highlighted this information in the revised version of the manuscript and discussed that our results suggest that the dysregulation of SPRYD7 is a common event in CRC not restricted to the KM12 cell model of CRC metastasis, and it is mostly associated to the metastatic capacity of CRC cell lines.
- The author examined the adhesion activity of CRC cells; however, high adhesion activity may inhibit the invasion capacity, which is not consistent with the conclusion. The author should show the images of Figure 2B in the manuscript or supplementary material.
As pointed out by the reviewer, during invasion, cell adhesions are not static as cells need to detach from the primary tumor. However, new cell adhesions between invasive cells and the extracellular matrix components of the basement membrane and from the metastatic niche are also formed, which indeed are mandatory for the colonization process. Thus, during invasion cell-cell adhesions are mainly disrupted but novel adhesions to the extracellular matrix and lymphatic and blood vessels are being continuously formed and detached, and thus, different cell adhesion patterns can be found at different time-points of the invasion process (Behrens 1993, Cavallaro and Christofori 2001, Schluter, Gassmann et al. 2006, Pramanik, Jolly et al. 2019). In the in vitro adhesion assay performed in this study, we analyzed the capacity of tumoral cells to adhere to the extracellular matrix (Matrigel matrix in our experiment). We found that Mock highly metastatic KM12SM and SW620 cells were more adherent than their isogenic poorly-metastatic KM12C and SW480 cells, respectively. In addition, SPRYD7 overexpression significantly increased the adhesion capacity of both low and high metastatic CRC cells, as well as the invasion capacity of these cells. Therefore, an increased invasion and adhesion capacity of these CRC cells after SPRYD7 overexpression should be in agreement with a more metastatic and tumorigenic phenotype of these cells.
Moreover, our belief is that highly metastatic KM12SM cells, which are more adherent cells than KM12C non-metastatic cells, showed more adhesive properties to get attached to the liver, as some literature also pointed out (Barderas et al. 2014, Luque-García et al. 2010, Jolly et al. 2019). Therefore, as SPRYD7 overexpression induced more adhesiveness, these increased properties should be able to help liver colonization, as also observed in the in vivo liver homing assays.
In this context, to address the concern of the reviewer, and as suggested also by Reviewer 2, we have modified Figure 2B to include the differences in the tumorigenic and metastatic properties of control isogenic CRC cells. A more aggressive phenotype of the KM12SM cells in comparison to the KM12C cells can be observed, which is in concordance with the results obtained from the in vitro assays after SPRYD7 overexpression. In addition, we have discussed the association between the adhesion and invasive capacity of tumoral cells in the revised version of the manuscript, including selected relevant references.
Furthermore, the original images of the colony soft-agar and wound healing assays have been included in the Supplementary Figure 3 of the revised version of the manuscript, as also suggested by Reviewer 4.
- In Figure 3, the author use 2 mice for Figure 3A and 3 mice for Figure 3B? The animal studies usually use 5 mice per group for statistical analyses.
Two different in vivo assays were performed to elucidate the role of SPRYD7 in CRC progression and metastasis.
On the one hand, we performed a homing assay where Mock and SPRYD7-stably transfected KM12C and KM12SM cells were intrasplenic injected in nude mice to analyze the effect of SPRYD7 in the migratory capacity to liver of CRC cells by PCR. This homing assay was performed to confirm an in vivo effect of SPRYD7 overexpression in the migratory and liver homing capacity of CRC cells, mainly in poorly-metastatic KM12C cells. To this end, two mice per group were used to reduce the number of experimental mice used, which is also based on previous bibliographic data of the group (Barderas, Mendes et al. 2013, Garranzo-Asensio, Solis-Fernandez et al. 2022, Solis-Fernandez, Montero-Calle et al. 2022, Barderas et al. 2012, Barderas et al. 2013). In this assay, a notably increment in the migratory and liver homing capacity of CRC cells was observed for SPRYD7 overexpressing KM12C cells, whereas SPRYD7 overexpression did not affect the ability of the highly metastatic to liver KM12SM cells. Remarkably, this in vivo assay was performed at 24 hours after intrasplenic cell injection, where only the presence or absence of human cells in the liver of nude mice was evaluated, and non-statistical analyses were performed.
On the other hand, for the tumor growth analysis, six mice were inoculated per cell line. In this experiment, after subcutaneous cell injection, tumors were allowed to grow for 21 days, and size and tumor volume monitored. At end point, representative images of the subcutaneous tumors (n=3) induced by Mock or SPRYD7 stably overexpressing cells were obtained. However, as depicted in Figure 3B, the statistical analyses indicated were performed with the six mice per group as well as the bar graph analyses of the tumor weight, and thus, increasing the reliability of the results.
To address the concern of the reviewer, and to avoid any misunderstanding, we have highlighted in the legend to the figure of Figure 3 of the revised version of the manuscript the number of animals used in each experiment. Additionally, we have also indicated that the images of the three tumors of each group were representative of each of the groups analyzed with the subcutaneous inoculation of Mock and SPRYD7 stably transfected cells.
- The employed proteomics analyses in Figure 5 and Figure 6; however, there are no relationship between these results, and there are no functional validation of these proteins, such as CRC proliferation and migration assays.
To address the concern of the reviewer, we have included in the discussion section of the revised version of the manuscript the relationship between the immunoprecipitation and the quantitative proteomics mass spectrometry analyses performed in this work. As expected, due to the different techniques and the different processing of the protein samples, a low correlation between both experiments was observed. Indeed, only 60 of the identified potential SPRYD7 interactors were also identified in the TMT quantitative proteomics analysis, and only 5 of them were found dysregulated due to SPRYD7 overexpression. Interestingly, three of the validated SPRYD7 interactors (SOD1, A2ML1, and FSCN1) were found dysregulated by SPRYD7 overexpression, which suggests an important role of these proteins in the SPRYD7-mediated role in CRC. This information has been highlighted in the revised version of the manuscript.
Moreover, as our main objective was to describe the effect of SPRYD7 in CRC progression and metastasis, as performed by the in vitro and in vivo functional assays, and the proteome and interactome associated to SPRYD7 upregulation in relation to CRC, we do believe that the functional validation of interacting proteins with SPRYD7 and dysregulated proteins by SPRYD7 is out of the scope of this manuscript, and although really interesting, this should be performed in another study involving the functional validation of proteins associated to SPRYD7. Therefore, to address the concern of the reviewers, we have acknowledged that the absence of the functional validation of validated proteins associated to SPRYD7 should be considered as a limitation of the study, and thus it has been included as such in the discussion section of the revised version of the manuscript.
- The author only performed SPRYD7 overexpression experiments in these CRC cells, the author should also design knockdown experiments in the cells that has a high expression level of SPRYD7.
In this manuscript, a plethora of different in vitro and in vivo experiments were performed with CRC cells overexpressing SPRYD7, which significantly confirmed the role of SPRYD7 in angiogenesis, and in CRC progression and metastasis. We agree with the reviewer that the knockdown of SPRYD7 in the highly metastatic KM12SM and SW620 cells CRC, with an endogenous higher SPRYD7 expression than their isogenic poorly-metastatic KM12C and SW480 CRC cells, respectively, could have been performed. However, we do believe that the results obtained with the overexpression of SPRYD7 are strong enough, and established in the literature to analyze the role of barely studied protein in CRC (i.e. Solís-Fernández et al. 2022 British Journal of Cancer), and other cancers (i.e. Zhu et al. 2017 Molecular Therapy Nucleic Acids).
In this context, to address the concern of the reviewer, we have emphasized in the discussion section of the revised version of the manuscript that although in this study the role of SPRYD7 in CRC has been widely confirmed by different analyses, the effect of SPRYD7 depletion in the highly metastatic CRC cells should also be performed to assess its role in CRC. As such, we have highlighted this limitation in the revised version of the manuscript.

Reviewer 2 Report
Summary and overall comments:
Authors have examined the role of SPRYD7 in colorectal cancer progression and metastasis. Montero-Calle et al show that SPRYD7 overexpression in non-metastatic cell line leads to gain in metastatic potential. Through various in vitro and in vivo studies, authors support the characterization of SPRYD7 in colorectal cancer.
The in vitro and in vivo studies show impressive results. Additionally, data from patients in TCGA also reveals a potential role of SPRYD7 in CRC prognosis. However, the proteomics data to identify potential SPRYD7 pathway and interactome are not conclusive. There are following major concerns to address:
Major comments:
1. For the phenotypic studies shown in Figure 2B, authors have normalized to the mock control of respective cell lines. How are the endogenous i.e non-overexpressing cells different from each other in their proliferation, wound healing, invasion, etc? Are endogenous KM12SM cells more invasive compared to KM12C cells?
2. The in vivo data shown in Figure 3 is highly intriguing. Were in vivo experiments performed with SW480 and SW620 cells as well? Since entire manuscript focuses in all four of these cell lines, was there any reason to omit in vivo validation for SW480 and SW620 cells?
3. It’s interesting that secretome from SW620 cells did not induce tube formation (Figure 4) even though that cell line is supposedly more metastatic compared to SW480 cells. Can authors comment on the observation?
4. Supplementary Figure 3: It looks like there are more than few hundred proteins i.e. each node/circle in the STRING plot. However, only 92 proteins were differentially expressed in the study. What was used as input for STRING analysis? Only the differentially expressed proteins should be used as input for STRING analysis. Additionally, for the protein enrichment analysis with DAVID, it will be more useful to show FDR of enrichment rather than number of proteins. Few Biological processed have high number of proteins, and thus number of proteins does not really represent any enrichment.
5. For the proteomics analysis, it is surprising that SPRYD7 is not one of the differentially expressed protein. Based on Supplementary Table S4, the protein was identified; however, the protein levels were similar with mock and SPRYD7 over-expressing cells. Why is there such discrepancy? We would expect high protein levels for SPRYD7 in SPRYD7 transfected cells. Were these cells not expressing SPRYD7? How stable was protein expression across the different assays that were performed?
6. As shown in Figure 5B, there is some overlap of similar proteomics changes between the KM12C and KM12SM cells. However, there are a good number of proteins that show opposing trend. Is there any biologically meaningful insights from these trends?
7. IP MS can result in several false positives, and thus it is essential to control for non-specific binding of proteins to antibodies as well as quantify the abundances of potential interactors. Using an isotype antibody as control for IP-MS, can authors confirm the absence of any of those 133 proteins? Additionally, only 8 proteins are common between the KM12C and KM12SM cells, which is highly concerning, and suggest that several proteins might be false positive. Using anti-ARG1 and anti-TYMP antibody IP in same cells, can authors confirm interaction of SPRYD7 with respective proteins?
8. Since cell lines have differential metastastic potential as well as SPRYD7 expressions, what was the rationale behind overexpression of SPRYD7 in cells? More discussion in intro and discussion will be helpful.
Minor comments:
1. Figure 6, STRING analysis plot is difficult to read.
Author Response
Reviewer 2
Comments and Suggestions for Authors
Summary and overall comments:
Authors have examined the role of SPRYD7 in colorectal cancer progression and metastasis. Montero-Calle et al show that SPRYD7 overexpression in non-metastatic cell line leads to gain in metastatic potential. Through various in vitro and in vivo studies, authors support the characterization of SPRYD7 in colorectal cancer.
The in vitro and in vivo studies show impressive results. Additionally, data from patients in TCGA also reveals a potential role of SPRYD7 in CRC prognosis. However, the proteomics data to identify potential SPRYD7 pathway and interactome are not conclusive.
We are very grateful to the reviewer for his/her work, for his/her comments, for the useful suggestions pointed out, and for the realization of the interest of the work.
There are following major concerns to address:
Major comments:
- For the phenotypic studies shown in Figure 2B, authors have normalized to the mock control of respective cell lines. How are the endogenous i.e non-overexpressing cells different from each other in their proliferation, wound healing, invasion, etc? Are endogenous KM12SM cells more invasive compared to KM12C cells?
To address the concern of the reviewer, we have modified Figure 2B to show the differences in the metastatic and tumorigenic capacities of tumoral cells between Mock cells and their corresponding isogenic poorly-metastatic KM12C and highly metastatic KM12SM cells (or SW480 and SW620 cells).
In the re-analysis it can be observed that the KM12SM highly metastatic cells were more proliferative, invasive, adherent, migratory, and with an increased colony formation capacity than their isogenic poorly-metastatic KM12C cells. In contrast, SW480 cells were more proliferative and migratory than their isogenic highly metastatic to lymphatic nodes SW620 cells, whereas no differences were observed in the invasive properties of these isogenic SW CRC cells. In addition, SW620 cells possessed a higher colony formation capacity than their isogenic non-metastatic SW480 cells.
To address the concern of the reviewer, we have modified Figure 2B, and we have discussed the differences observed between low and high metastatic Mock cells in the revised version of the manuscript.
- The in vivo data shown in Figure 3 is highly intriguing. Were in vivo experiments performed with SW480 and SW620 cells as well? Since entire manuscript focuses in all four of these cell lines, was there any reason to omit in vivo validation for SW480 and SW620 cells?
Thank you to the reviewer for his/her comment. In response to the reviewer, we performed the in vivo analyses on the KM12 cell model of liver metastasis to analyze not only liver homing but also the tumor growth. Although we did perform the in vitro analyses using all four cell lines, we only focused in the in vivo assays on KM12C and KM12SM cells for this reason and to reduce the number of experimental animals used in the study. Since, with these experiments we could validate the effect of SPRYD7 in CRC progression and metastasis in vivo, we have maintained the in vivo results as previously observed in the original version of the manuscript.
In this context, to address the concern of the reviewer, we have included this information in the revised version of the manuscript.
- It’s interesting that secretome from SW620 cells did not induce tube formation (Figure 4) even though that cell line is supposedly more metastatic compared to SW480 cells. Can authors comment on the observation?
To address the concern of the reviewer, we have discussed in detail the results derived from the tube formation assay regarding SW620 cells. SW620 cells are more metastatic compared to SW480 cells, and in our hands showed lower tube formation ability than SW480 cells that could be associated to the metastatic nature of SW620 cells. SW620 cells metastasized to lymph nodes and probably they do not need to form more tubes than SW480 cells to proliferate, migrate and colonize other organs.
Therefore, we have discussed this information in the revised version of the manuscript.
- Supplementary Figure 3: It looks like there are more than few hundred proteins i.e. each node/circle in the STRING plot. However, only 92 proteins were differentially expressed in the study. What was used as input for STRING analysis? Only the differentially expressed proteins should be used as input for STRING analysis. Additionally, for the protein enrichment analysis with DAVID, it will be more useful to show FDR of enrichment rather than number of proteins. Few Biological processed have high number of proteins, and thus number of proteins does not really represent any enrichment.
We thank the reviewer for his/her comment that has significantly improved the aesthetic of the Supplementary Figure 3 (Supplementary Figure 4 in the revised version of the manuscript). An old version of the STRING and Gene ontology analyses was shown in the supplementary figure, in which more than the 92 dysregulated protein were represented. As suggested by the reviewer, we have modified the supplementary figure to show only the pathways in which the 92 dysregulated proteins are involved. We have also modified the GO graphs to include the significantly enriched pathways in which more proteins dysregulated by SPRYD7 overexpression are involved (FDR < 0.05).
- For the proteomics analysis, it is surprising that SPRYD7 is not one of the differentially expressed protein. Based on Supplementary Table S4, the protein was identified; however, the protein levels were similar with mock and SPRYD7 over-expressing cells. Why is there such discrepancy? We would expect high protein levels for SPRYD7 in SPRYD7 transfected cells. Were these cells not expressing SPRYD7? How stable was protein expression across the different assays that were performed?
We used as control of the overexpression of SPRYD7 the mRNA levels assessed by qPCR and protein levels assessed by WB prior to further experiments. Additionally, we performed these assays prior to proteomics analyses. As indicated by the reviewer, we did not observe overexpression of SPRYD7 by mass spectrometry in KM12C cells (1.01 ratio SPRYD7/Mock KM12C cells) or KM12SM cells (1.11 ratio SPRYD7/Mock KM12SM cells). We believe that this could be associated to the known phenomenon referred to as ratio compression, where distinct TMT labeled peptide ions with close precursor m/z values could be co-isolated and co-fragmented in MS2, skewing reporter ion intensities towards a 1:1 ratio (Savitski et al. 2013, Ting et al. 2011). Moreover, as we also observed by principal component analysis a separation between Mock and SPRYD7 cells, and we observed by qPCR and WB a correct overexpression of SPRYD7, we believe that the ratio compression would be behind the discrepancy indicated by the reviewer.
Therefore, to address the concern of the reviewer, we have included this information in the revised version of the manuscript, with appropriated references supporting the ratio compression phenomenon.
- As shown in Figure 5B, there is some overlap of similar proteomics changes between the KM12C and KM12SM cells. However, there are a good number of proteins that show opposing trend. Is there any biologically meaningful insights from these trend.
We do believe that these changes could be associated to the different metastatic properties of KM12 cells and the endogenous levels of SPRYD7, where most in vitro and in vivo effects were more noticeable in the KM12C cells in comparison to the KM12SM highly metastatic cells. Therefore, to address the concern of the reviewer we have discussed this information in the revised version of the manuscript.
- IP MS can result in several false positives, and thus it is essential to control for non-specific binding of proteins to antibodies as well as quantify the abundances of potential interactors. Using an isotype antibody as control for IP-MS, can authors confirm the absence of any of those 133 proteins? Additionally, only 8 proteins are common between the KM12C and KM12SM cells, which is highly concerning, and suggest that several proteins might be false positive. Using anti-ARG1 and anti-TYMP antibody IP in same cells, can authors confirm interaction of SPRYD7 with respective proteins?
Thank you to the reviewer for his/her comment. We used the proteins non-specifically anchored to the beads prior to the incubation with the beads and SPRYD7 antibody as negative control of the immunoprecipitation assays to remove from the analysis non-specific binding proteins. In addition, proteins that appear in more than the 15% of the immunoaffinity assays registered in the CRAPome data base were also removed from the analysis to avoid false positive. In this way, we might be losing true SPRYD7 interacting proteins that are also non-specifically anchored to the beads, but we enrich the list of candidate proteins in actual SPRYD7 interactors.
Regarding the low correlation between the KM12C and KM12SM IPs, it might be associated to a non-detected bias in the sample processing and mass spectrometry analysis, as each IP was performed and analyzed separately, but most probably because of the differences in the metastatic properties of the cells that would provide differences in the dysregulated proteins associated to SPRYD7 or its interacting proteins. Therefore, for consistency, we used the mass spectrometry analysis as a tool for the identification of potential novel SPRYD7 interactors, but candidate proteins identified from one or the two IPs were then validated by western blot using the four CRC cell lines overexpressing SPRYD7 for further confirmation of their role as actual SPRYD7 interactors. In addition, to address the concern of the reviewer we wanted to focus our cross-validation analyses on TYMP, TOLLIP, ARG1 and IL1RN. However, since the antibodies for TYMP and TOLLIP were not recommended for IP analyses, we finally focused on ARG1 and IL1RN, confirming their interaction with SPRYD7 by performing a cross-immunoprecipitation. In this experiment, we could find SPRYD7 among ARG1, and IL1RN interactors by western blot.
In this context, to fix this issue we have included these new data as a new panel C in Figure 6. Additionally, we have discussed the new results in the results section of the revised version of the manuscript.
- Since cell lines have differential metastastic potential as well as SPRYD7 expressions, what was the rationale behind overexpression of SPRYD7 in cells? More discussion in intro and discussion will be helpful.
We have carefully revised the manuscript to emphasize the relevance of SPRYD7 overexpression in isogenic CRC cell lines with different metastatic abilities.
Remarkably, the rationale behind the overexpression of SPRYD7 in SW480, SW620, KM12C, and KM12SM cells was to try to determine whether the overexpression of SPRYD7 would increase the tumorigenic and metastatic properties of these cells. Additionally, the rationale behind the use of KM12C and KM12SM CRC cells for the in vivo assays was to survey for changes in liver homing metastasis and tumor growth and determine whether SPRYD7 could induce poorly-metastatic cells to become metastatic or able to colonize the liver.
Therefore, to address the concern of the reviewer, we have included this information in the revised version of the introduction section of the manuscript.
Minor comments:
- Figure 6, STRING analysis plot is difficult to read.
To address the concern of the reviewer, we have modified the STRING analysis of Figure 6 to improve the reading of the figure by increasing the label sizes.

Reviewer 3 Report
The manuscript investigates the role of SPRYD7, a relatively unknown protein, in the progression and metastasis of colorectal cancer (CRC). The study begins by demonstrating the upregulation of SPRYD7 in highly metastatic CRC cells and its association with poor prognosis in patients. In vitro and in vivo experiments reveal that CRC cells overexpressing SPRYD7 exhibit enhanced invasive, migratory, and anchorage-independent growth capabilities compared to control cells. Moreover, SPRYD7 promotes liver homing and tumor growth in vivo, with tumors derived from SPRYD7-overexpressing cells showing increased vascularization and elevated levels of angiogenic markers. Proteomic analysis of SPRYD7 overexpression identifies dysregulated biological processes related to cell cycle, DNA replication, RNA processing, cytoskeleton regulation, and cell junctions. Interactome analysis reveals proteins interacting with SPRYD7, which are involved in immune and inflammatory responses, cell adhesion, and actin cytoskeleton regulation. Overall, the findings support the role of SPRYD7 in promoting invasion, adhesion, migration, and angiogenesis in CRC. The study highlights SPRYD7 as a potential therapeutic target and provides insights into the underlying pathological mechanisms of CRC progression and metastasis.
To improve the manuscript, the following suggestions could be considered:
1) As mentioned in the Discussion part, the authors could explore the effects of inhibiting SPRYD7 expression or activity in CRC cells. This could be achieved through genetic knockdown or the use of pharmacological inhibitors. Assess the impact of SPRYD7 inhibition on cellular phenotypes, including migration, invasion, and angiogenesis.
2) The authors could investigate the potential synergistic effects of targeting SPRYD7 in combination with existing CRC therapies, such as chemotherapy or targeted therapies. Assess whether SPRYD7 inhibition enhances the efficacy of standard treatments and reduces the metastatic potential of CRC cells.
3) It would be more meaningful to perform Protein dysregulation profiling in tumors from Results 3.3 instead of CRC cells.
Minor revision:
2. Figure 3A, “pb”, should be “bp”
Author Response
Reviewer 3
Comments and Suggestions for Authors
The manuscript investigates the role of SPRYD7, a relatively unknown protein, in the progression and metastasis of colorectal cancer (CRC). The study begins by demonstrating the upregulation of SPRYD7 in highly metastatic CRC cells and its association with poor prognosis in patients. In vitro and in vivo experiments reveal that CRC cells overexpressing SPRYD7 exhibit enhanced invasive, migratory, and anchorage-independent growth capabilities compared to control cells. Moreover, SPRYD7 promotes liver homing and tumor growth in vivo, with tumors derived from SPRYD7-overexpressing cells showing increased vascularization and elevated levels of angiogenic markers. Proteomic analysis of SPRYD7 overexpression identifies dysregulated biological processes related to cell cycle, DNA replication, RNA processing, cytoskeleton regulation, and cell junctions. Interactome analysis reveals proteins interacting with SPRYD7, which are involved in immune and inflammatory responses, cell adhesion, and actin cytoskeleton regulation. Overall, the findings support the role of SPRYD7 in promoting invasion, adhesion, migration, and angiogenesis in CRC. The study highlights SPRYD7 as a potential therapeutic target and provides insights into the underlying pathological mechanisms of CRC progression and metastasis.
We are very grateful to the reviewer for his/her work, for his/her comments, for the useful suggestions pointed out, and for the realization of the interest of the work.
To improve the manuscript, the following suggestions could be considered:
- As mentioned in the Discussion part, the authors could explore the effects of inhibiting SPRYD7 expression or activity in CRC cells. This could be achieved through genetic knockdown or the use of pharmacological inhibitors. Assess the impact of SPRYD7 inhibition on cellular phenotypes, including migration, invasion, and angiogenesis.
We thank the reviewer for his/her comments. We agree with the reviewer that the knockdown of SPRYD7 in the highly metastatic KM12SM and SW620 cells CRC, with an endogenous higher SPRYD7 expression than their isogenic poorly-metastatic KM12C and non-metastatic SW480 CRC cells, respectively, would increase the reliability of the results, and thus, it has been indicated as a limitation of the study in the discussion section of the manuscript. However, different in vitro and in vivo experiments with the CRC cells overexpressing SPRYD7 have been performed in this study which significantly confirm the role of SPRYD7 in angiogenesis, and in CRC progression and metastasis.
To address the concern of the reviewer, we have emphasized in the discussion section of the revised version of the manuscript that although the role of SPRYD7 in CRC have been widely confirmed by different analyses in this study, the effect of SPRYD7 depletion in the highly metastatic CRC cells is mandatory to firmly confirm its role in CRC.
- The authors could investigate the potential synergistic effects of targeting SPRYD7 in combination with existing CRC therapies, such as chemotherapy or targeted therapies. Assess whether SPRYD7 inhibition enhances the efficacy of standard treatments and reduces the metastatic potential of CRC cells.
- It would be more meaningful to perform Protein dysregulation profiling in tumors from Results 3.3 instead of CRC cells.
We thank the reviewer for these comments. However, we believe that their study would be enough to compose another highly relevant manuscript. Therefore, to address the concern of the reviewer we have acknowledged these comments as part of the future research involving SPRYD7 to determine its usefulness as therapeutic target of CRC and CRC metastasis.
Minor revision:
- Figure 3A, “pb”, should be “bp”
Thanks to the comment of the reviewer, we have realized about the presence of this typo in the original version of the manuscript. We apologize for the mistake.
We have fixed this issue in the revised version of Figure 3A.

Reviewer 4 Report
The manuscript is well-written. The experiments are properly designed. I have a few suggestions as below.
In figure 2, I suggest authors to provide the original representative images other than the quantification. For example, Colony soft agar assay and wound healing assay. These may provide more direct idea for readers.
Author Response
Reviewer 4
Comments and Suggestions for Authors
The manuscript is well-written. The experiments are properly designed. I have a few suggestions as below.
We are very grateful to the reviewer for his/her work, for his/her suggestion and for the realization of the interest of the work.
- In figure 2, I suggest authors to provide the original representative images other than the quantification. For example, Colony soft agar assay and wound healing assay. These may provide more direct idea for readers.
To address the concern of the reviewer, the original images of the colony soft agar and wound healing assays have been included as Supplementary Figure 3 in the revision version of the manuscript, as also requested by Reviewer #1.

Round 2
Reviewer 2 Report
Authors have addressed comments and concerns. Clarity of figures can still be improved - not sure if it's an issue while importing in PDF.
Author Response
Reviewer 2
Comments and Suggestions for Authors
Authors have addressed comments and concerns. Clarity of figures can still be improved - not sure if it's an issue while importing in PDF.
We are very grateful to the reviewer for his/her work, for his/her comments, and for the useful suggestions pointed out during the peer review to improve the manuscript.
We have improved the quality and resolution of the figures in the revised version of the manuscript.

Reviewer 3 Report
The authors have addressed all my concerns and I have no further question.